



# Inter-comparison of $O_2/N_2$ Ratio Scales Among AIST, NIES, TU, and SIO Based on Round-Robin Using Gravimetric Standard Mixtures

Nobuyuki Aoki[1], Shigeyuki Ishidoya[2], Yasunori Tohjima[3], Shinji Morimoto[4], Ralph F. Keeling[5], Adam Cox[5], Shuichiro Takebayashi[4] and Shohei Murayama[2]

[1]National Metrology Institute of Japan (NMIJ), National Institute of Advanced Industrial Science and Technology (AIST), 1-1-1 Umezono, Tsukuba 305-8563, Japan
[2]Research Institute for Environmental Management Technology (EMRI), National Institute of Advanced Industrial Science and Technology (AIST), Tsukuba 305-8569, Japan
[3]Center for Environmental Measurement and Analysis, National Institute for Environmental Studies, Tsukuba 305-8506, Japan
[4]Center for Atmospheric and Oceanic Studies, Graduate School of Science, Tohoku University, Sendai 980-8578, Japan
[5]Scripps Institution of Oceanography, La Jolla, CA 92093-0244, USA

*Correspondence to*: Nobuyuki Aoki (aoki-nobu@aist.go.jp)

**Abstract.** A study was conducted to compare the $\delta(O_2/N_2)$ scales used by four laboratories engaged in atmospheric $\delta(O_2/N_2)$ measurements. These laboratories are the Research Institute for Environmental Management Technology, Advanced Industrial Science and Technology (EMRI/AIST), the National Institute for Environmental Studies (NIES), Tohoku University (TU), and Scripps Institution of Oceanography (SIO). Therefore, five high-precision standard mixtures for $O_2$ molar fraction gravimetrically prepared by the National Metrology Institute of Japan (NMIJ), AIST (NMIJ/AIST) with a standard uncertainty of less than 5 per meg were used as round-robin standard mixtures. EMRI/AIST, NIES, TU, and SIO reported the analysed values of the standard mixtures on their own $\delta(O_2/N_2)$ scales, and the values were compared with the $\delta(O_2/N_2)$ values gravimetrically determined by NMIJ/AIST (the NMIJ/AIST scale). The $\delta(O_2/N_2)$ temporal drift in the five standard mixtures during the inter-comparison experiment was corrected based on the $\delta(O_2/N_2)$ values analysed before and after the experiments by EMRI/AIST. The scales are compared based on offsets in zero and span. The span offsets from the NMIJ/AIST scale ranged from −0.17% to 3.3%, corresponding with the difference of 0.29 Pg yr$^{-1}$ in the estimates for land biospheric and oceanic $CO_2$ uptakes. The zero offsets from the NMIJ/AIST scale are −581.0 ± 2.2, −221.4 ± 3.1, −243.0 ± 3.0, and −50.7 ± 2.4 per meg for EMRI/AIST, TU, NIES, and SIO, respectively. The atmospheric $\delta(O_2/N_2)$ values observed at Hateruma Island (HAT; 24.05°N, 123.81°E), Japan, by EMRI/AIST and NIES became comparable by converting their scales to the NMIJ/AIST scale.



## 1. Introduction

Observing the long-term change in atmospheric $O_2$ molar fraction, combined with $CO_2$ observation, enables us to estimate terrestrial biospheric and oceanic $CO_2$ uptakes separately. $O_2$ is exchanged with $CO_2$ with some stoichiometric ratios for terrestrial biospheric activities and fossil fuel combustion. Meanwhile, the ocean $CO_2$ uptake and $O_2$ emissions are decoupled

since the ocean acts as a carbon sink by physiochemically dissolving the $CO_2$ (e.g., Keeling et al., 1993). Various laboratories have performed changes in atmospheric $O_2$ since the early 1990s (e.g., Keeling et al., 1996; Bender et al., 2005; Manning and Keeling, 2006; Tohjima et al., 2008, 2019; Ishidoya et al., 2012a, b; Goto et al., 2017). Recently, Resplandy et al. (2019) introduced a method to estimate the global ocean heat content (OHC) increase based on atmospheric $O_2$ and $CO_2$ measurements. They extracted solubility-driven components of the atmospheric potential oxygen ($APO = O_2 + 1.1 \times CO_2$) (Stephens et al.,

1998) by combining their observational results with climate and ocean models. The global OHC change is a fundamental measure of global warming. Indeed, the ocean uptakes more than 90% of the earth's excess energy and is evaluated based on ocean temperature measurements using Argo float (e.g., Levitus et al., 2012). Thus, the atmospheric $O_2$ measurements are linked to the global $CO_2$ budget and OHC.

The approaches described above rely on precision measurements that can detect micro-mole-per-mole-level changes in

atmospheric $O_2$ molar fraction (~21%). After Keeling and Shertz (1992) succeeded in developing the measurement technique based on the interferometer, various measurement techniques have been developed to quantify atmospheric $O_2$ molar fraction, including using mass spectrometry (Bender et al., 1994; Ishidoya et al., 2003; Ishidoya and Murayama, 2014), a paramagnetic technique (Manning et al., 1999; Ishidoya et al., 2017; Aoki and Shimosaka, 2018), a vacuum-ultraviolet absorption technique (Stephens et al., 2003), gas chromatography (Tohjima, 2000), a method using fuel cells (Stephens et al., 2007; Goto et al.,

2013), and cavity ring-down spectroscopy analyser (Berhanu et al., 2019). All programs have reported changes in $O_2$ regarding the equivalent changes in the $O_2/N_2$ ratio by convention. This is expressed as the relative change compared to an arbitrary reference (Keeling and Shertz, 1992; Keeling et al., 2004) in per meg (one per meg is equal to $1 \times 10^{-6}$).

$$\delta(O_2/N_2) = \frac{[n(O_2)/n(N_2)]_{sam}}{[n(O_2)/n(N_2)]_{ref}} - 1 \tag{1}$$

In the equation, $n$ depicts the molar amount of each substance, and the subscripts sam and ref represent sample and reference air, respectively. The $\delta(O_2/N_2)$ value multiplied by $10^6$ is expressed in per meg units. The $O_2$ molar fractions in air are 20.946% (Machta and Hughes, 1970). Therefore, adding 1 µmol of $O_2$ to a mole of dry air will increase in $\delta(O_2/N_2)$ by 4.8 per meg. Each laboratory has typically employed its own $O_2/N_2$ reference based on natural air compressed and stored in high-pressure

cylinders. Each laboratory has also assumed responsibility for calibrating the relationship between the measured instrument response and the reported change per meg units (span sensitivity). Therefore, the reported trends in $O_2/N_2$ are potentially biased by any long-term drift in the $O_2/N_2$ ratio of the reference cylinders (zero drift) or errors in the assumed span sensitivity the instrument (span error). Note that an uncertainty below 5 per meg is required for the global $CO_2$ budget analyses based on



$\delta(O_2/N_2)$ observations [Table 2 in Keeling et al. (1993)]. Challenges in achieving this precision include fractionations of $O_2$ and $N_2$ induced by pressure, temperature, and water vapour gradients (Keeling et al., 2007), adsorption/desorption of the constituents on the cylinder's inner surface (Leuenberger et al., 2015), and permeation/leakage of the constituents from/through the valve (Sturm et al., 2004; Keeling et al., 2007). Tohjima et al. (2005) developed high-precision $O_2$ standard mixtures with

15 per meg uncertainty for $\delta(O_2/N_2)$ to resolve these problems by preparing gravimetric standard mixtures of pure $N_2$, $O_2$, Ar, and $CO_2$. Their study was significant, but the uncertainties remain larger than those recommended by Keeling et al. (1993), as mentioned above.

Recently, a technique was developed for preparing high-precision primary standard mixtures with standard uncertainties less than 5 per meg for $\delta(O_2/N_2)$ at the National Metrology Institute of Japan, National Institute of Advanced Industrial Science

and Technology (NMIJ/AIST) (Aoki et al., 2019). The high-precision standard mixtures allow us to evaluate scale zero and span offsets accurately and precisely. In this study, we conducted inter-comparison experiments to compare span sensitivities among the $O_2/N_2$ scales of Research Institute for Environmental Management Technology, Advanced Industrial Science and Technology (EMRI/AIST), National Institute for Environmental Studies (NIES), Tohoku University (TU), and Scripps Institution of Oceanography (SIO) using the developed high-precision standard mixtures. Following this, a regression analysis

is applied to the inter-comparison results to investigate the relationship between the individual laboratory $O_2/N_2$ scales. Results showed a slight but significant difference in the span sensitivities of the individual scales. Finally, we compare the atmospheric $\delta(O_2/N_2)$ values observed on the EMRI/AIST scale with those on the NIES scale for the air samples collected at Hateruma Island (HAT; 24°03'N, 123°49'E), Japan, using the relationship between the individual laboratory scales obtained in this study.

## 2. Experimental Procedures

### 2.1 NMIJ/AIST Scale and Round-Robin Standard Mixtures

In this study, five high-precision standard mixtures with standard uncertainties less than 5 per meg for $\delta(O_2/N_2)$ were used as round-robin standard mixtures. The NMIJ/AIST previously mixed them gravimetrically following ISO 6142-1:2015 (Aoki et al., 2019), which were contained in 10 L aluminium-alloy cylinders (Luxfer Gas Cylinders, UK) with a diaphragm valve (G-55, Hamai Industries Limited, Japan). Table 1 shows the gravimetrically determined molar fractions for $N_2$, $O_2$, Ar, $CO_2$, as

well as $\delta(O_2/N_2)$ in the round-robin mixtures. However, the gravimetric values of $N_2$, $O_2$, Ar, and $CO_2$ molar fractions were recalculated based on the cylinders' updated expansion rate. The value was determined as $1.62 \pm 0.06$ ml $Mpa^{-1}$ (unpublished data), which was determined by measuring expansion volume of a cylinder with an increase of inner pressure of the cylinders sunk in water since the previous expansion rate ($2.2 \pm 0.2$ ml $Mpa^{-1}$) was provided by a cylinder supplier. The source gases used are pure $CO_2$ (>99.998%, Nippon Ekitan Corp., Japan), pure Ar (99.9999%, G1-grade, Japan Fine Products, Japan), pure

$O_2$ (99.99995%, G1-grade, Japan Fine Products, Japan), and pure $N_2$ (99.99995%, G1-grade, Japan Fine Products, Japan). Impurities in the source gases were identified and quantified via several techniques, including gas chromatography (GC). GC



equipped with a thermal conductivity detector (GC/TCD) was used to analyse $N_2$, $O_2$, $CH_4$, and $H_2$ in pure $CO_2$. $O_2$ and Ar in pure $N_2$ and $N_2$ in pure $O_2$ were analysed using GC, equipped with a mass spectrometer. A Fourier-transform infrared spectrometer was used to detect $CO_2$, $CH_4$, and CO in pure $N_2$, $O_2$, and Ar. A galvanic cell $O_2$ analyser was used to quantify $O_2$ in pure Ar. A capacitance-type moisture sensor measured $H_2O$ in pure $CO_2$, and a cavity ring-down moisture analyser

measured $H_2O$ in pure $N_2$, $O_2$, and Ar.

In this study, the absolute $O_2/N_2$ scale determined using the round-robin standard mixtures is hereafter the NMIJ/AIST scale. The NMIJ/AIST scale is presented only for scientific research and is uncertified by NMIJ. Here, $\delta(O_2/N_2)_{NMIJ/AIST}$ represents the $\delta(O_2/N_2)$ on the NMIJ/AIST scale, which was calculated against a reference $O_2/N_2$ ratio of $0.20946/0.78084 = 0.26825$, previously reported (Machta and Hughes, 1970). The range of $\delta(O_2/N_2)_{NMIJ/AIST}$ values for the round-robin standard mixtures

was −4200 per meg to 2200 per meg. The standard uncertainties of the $\delta(O_2/N_2)_{NMIJ/AIST}$ values were 3.3 per meg to 4.0 per meg.

**2.2 Procedure of Inter-comparison**

The EMRI/AIST, NIES, TU, and SIO conducted the inter-comparison experiment. Five round-robin standard mixtures were analysed in the order of EMRI/AIST (May to July 2017), NIES (September to November 2017), TU (December 2017 to

January 2018), and SIO (May to December 2018). Each lab reported the $\delta(O_2/N_2)_{round\text{-}robin}$ values were determined against their scales to the NMIJ/AIST. The subscript round-robin is hereafter the round-robin standard mixture. Each lab analysed air delivered from the cylinders after placing them horizontally for more than five days after their transport to avoid the change of $\delta(O_2/N_2)_{round\text{-}robin}$ values in the standard mixtures by thermal diffusion and gravitational fractionation. The $\delta(O_2/N_2)_{round\text{-}robin}$ values determined by individual laboratories using their methods were compared with the $\delta(O_2/N_2)_{NMIJ/AIST}$ values.

EMRI/AIST and TU used mass spectrometry, NIES used GC, and SIO used the interferometric method, as summarised in Table 2. The stability of $O_2/N_2$ ratios in the round-robin standard mixtures during the inter-comparison experiment was evaluated by analysing their $\delta(O_2/N_2)_{round\text{-}robin}$ values using a mass spectrometer (Delta-V, Thermo Fisher Scientific Inc., USA) (Ishidoya and Murayama, 2014) at EMRI/AIST before and after the inter-comparison experiment.

Ar molar fractions in the round-robin standard mixtures were from 9297 to 9351 $\mu mol\ mol^{-1}$, much more variable than

variations in the tropospheric air (less than 1 $\mu mol\ mol^{-1}$) (Keeling et al., 2004). Isotopic ratios of $\delta(^{17}O/^{16}O)$, $\delta(^{18}O/^{16}O)$, and $\delta(^{15}N/^{14}N)$ in the round-robin standard mixtures, measured using the mass spectrometer by EMRI/AIST, were lower than the atmospheric values by 4.7‰, 9‰, and 2.4‰, respectively. $\delta(^{17}O/^{16}O)$, $\delta(^{18}O/^{16}O)$, and $\delta(^{15}N/^{14}N)$ are expressed as

$$\delta\left(^{17}O/^{16}O\right) = \frac{[n(^{17}O)/n(^{16}O)]_{sam}}{[n(^{17}O)/n(^{16}O)]_{ref}} - 1 \tag{2}$$

$$\delta\left(^{18}O/^{16}O\right) = \frac{[n(^{18}O)/n(^{16}O)]_{sam}}{[n(^{18}O)/n(^{16}O)]_{ref}} - 1 \tag{3}$$


$$\delta\left(^{15}N/^{14}N\right) = \frac{[n(^{15}N)/n(^{14}N)]_{sam}}{[n(^{15}N)/n(^{14}N)]_{ref}} - 1. \tag{4}$$

Here, the isotopic ratios of $\delta(^{17}O/^{16}O)$, $\delta(^{18}O/^{16}O)$, and $\delta(^{15}N/^{14}N)$ were approximately equal to those of $\delta(^{17}O^{16}O/^{16}O^{16}O)$, $\delta(^{18}O^{16}O/^{16}O^{16}O)$, and $\delta(^{15}N^{14}N/^{14}N^{14}N)$. This is because $^{17}O^{17}O/^{16}O^{16}O$, $^{18}O^{18}O/^{16}O^{16}O$, and $^{15}N^{15}N/^{14}N^{14}N$ tended to be lower than $^{17}O^{16}O/^{16}O^{16}O$, $^{18}O^{16}O/^{16}O^{16}O$, and $^{15}N^{14}N/^{14}N^{14}N$ by 5000 times, 1000 times, and 500 times, respectively.

We applied the following corrections to the measured $\delta(O_2/N_2)_{round-robin}$ values from the individual laboratories by considering the deviations of Ar molar fraction and isotopic ratios in the round-robin standard mixtures from the tropospheric air. The $\delta(O_2/N_2)_{round-robin}$ values reported by EMRI/AIST and TU were corrected based on the deviation in the isotope ratio from the atmospheric level using isotopic ratios of N and O measured simultaneously at EMRI/AIST. This is because they measured the values of $\delta(^{16}O^{16}O/^{14}N^{14}N)$ and $\delta(^{16}O^{16}O/^{14}N^{15}N)$, respectively. NIES corrected the Ar molar fraction difference from its atmospheric level since the $O_2$ peak obtained in GC included the Ar peak. SIO also corrected the difference in the Ar molar fraction using the round-robin standard mixtures' gravimetric values since they only measured $O_2$ molar fractions. The measurement techniques and calculation procedures of the $\delta(O_2/N_2)_{round-robin}$ values for individual laboratories are detailed in the next section.

## 2.3 Analytical and Calculation Methods of $\delta(O_2/N_2)$ Values

### 2.3.1 EMRI/AIST

The $\delta(O_2/N_2)_{round-robin}$ values for EMRI/AIST were calculated based on the $\delta(^{16}O^{16}O/^{14}N^{14}N)_{round-robin}$ values measured using the mass spectrometer. The $\delta(^{16}O^{16}O/^{14}N^{14}N)_{round-robin}$ values were calculated against the reference air on the EMRI/AIST scale, which is natural air filled in a 48 L aluminium cylinder with a diaphragm valve (G-55, Hamai Industries Limited, Japan). The measurement technique's detail was given in Ishidoya and Murayama (2014). The mass spectrometer was adjusted to measure ion beam currents for masses 28 ($^{14}N^{14}N$), 29 ($^{15}N^{14}N$), 32 ($^{16}O^{16}O$), 33 ($^{17}O^{16}O$), 34 ($^{18}O^{16}O$), and 44 ($^{12}C^{16}O^{16}O$). The $\delta(O_2/N_2)_{NMIJ/AIST}$ in the round-robin standard mixtures comprising all isotopes of $O_2$ and $N_2$ are unequal to the isotopic ratios of $\delta(^{16}O^{16}O/^{14}N^{14}N)_{round-robin}$ measured using the mass spectrometer. Thus, mass-spectrometry-based isotopic ratios must be converted to values equivalent to the $\delta(O_2/N_2)_{NMIJ/AIST}$ values. The $\delta(O_2/N_2)_{round-robin}$ values were calculated based on isotopic ratios $^{15}N^{14}N/^{14}N^{14}N$, $^{17}O^{16}O/^{16}O^{16}O$, and $^{18}O^{16}O/^{16}O^{16}O$ in the round-robin standard mixtures and reference air, as shown in Eq. (5).

$$\delta(O_2/N_2)_{round-robin} = \left[\delta(^{16}O^{16}O/\,^{14}N\,^{14}N) + 1\right]_{round-robin} \times$$
$$\left[\frac{1+^{17}O^{16}O/^{16}O^{16}O+^{18}O^{16}O/^{16}O^{16}O}{1+^{15}N\,^{14}N/^{14}N^{14}N}\right]_{round-robin} \bigg/ \left[\frac{1+^{17}O^{16}O/^{16}O^{16}O+^{18}O^{16}O/^{16}O^{16}O}{1+^{15}N\,^{14}N/^{14}N\,^{14}N}\right]_{ref} - 1. \qquad (5)$$

Here, isotopic species of $^{17}O^{17}O$, $^{18}O^{17}O$, $^{18}O^{18}O$, and $^{15}N^{15}N$ were negligible since their abundance was sufficiently small. The isotopic ratios of $^{15}N^{14}N/^{14}N^{14}N$, $^{17}O^{16}O/^{16}O^{16}O$, and $^{18}O^{16}O/^{16}O^{16}O$ in the round-robin standard mixtures were calculated using Eqs. (6), (7), and (8).





$$^{18}O^{16}O/^{16}O^{16}O = [\delta(^{18}O^{16}O/^{16}O^{16}O)_{round\text{-}robin} + 1] \times (^{18}O^{16}O/^{16}O^{16}O)_{ref}, \tag{6}$$

$$^{17}O^{16}O/^{16}O^{16}O = [\delta(^{17}O^{16}O/^{16}O^{16}O)_{round\text{-}robin} + 1] \times (^{17}O^{16}O/^{16}O^{16}O)_{ref}, \tag{7}$$

$$^{15}N^{14}N/^{14}N^{14}N = [\delta(^{15}N^{14}N/^{14}N^{14}N)_{round\text{-}robin} + 1] \times (^{15}N^{14}N/^{14}N^{14}N)_{ref}. \tag{8}$$

The isotopic ratios of $\delta(^{15}N^{14}N/^{14}N^{14}N)_{round\text{-}robin}$, $\delta(^{17}O^{16}O/^{16}O^{16}O)_{round\text{-}robin}$, and $\delta(^{18}O^{16}O/^{16}O^{16}O)_{round\text{-}robin}$ were determined against the EMRI/AIST reference air. Values of $(^{18}O^{16}O/^{16}O^{16}O)_{ref}$, $(^{17}O^{16}O/^{16}O^{16}O)_{ref}$, and $(^{15}N^{14}N/^{14}N^{14}N)_{ref}$ refer to ratios of $^{18}O^{16}O/^{16}O^{16}O$, $^{17}O^{16}O/^{16}O^{16}O$, and $^{15}N^{14}N/^{14}N^{14}N$ in the reference air. We regard the isotopic ratios in the EMRI/AIST

reference air as atmospheric values since differences between $N_2$, $O_2$, and Ar in the AIST reference air and air samples at Hateruma were small enough to be negligible. Therefore, the corresponding atmospheric values were used to calculate the ratios of $(^{18}O^{16}O/^{16}O^{16}O)_{ref}$, $(^{17}O^{16}O/^{16}O^{16}O)_{ref}$, and $(^{15}N^{14}N/^{14}N^{14}N)_{ref}$, since isotopic abundances in the troposphere are constant (Junk and Svec, 1958; Baertschi, 1976; Li et al., 1988; Barkan and Luz, 2005).

**2.3.2 NIES**

NIES reported the $\delta(O_2/N_2)_{round\text{-}robin}$ values based on the $\delta\{(O_2+Ar)/N_2\}_{round\text{-}robin}$ values measured using a GC/TCD (Tohjima, 2000). The $\delta\{(O_2+Ar)/N_2\}_{round\text{-}robin}$ values were calculated against the reference air on the NIES scale, which is natural air filled in a 48 L aluminium cylinder. A column separates the $(O_2 + Ar)$ and $N_2$ in the air sample, and a TCD detected the individual peaks. The reference and sample air were repeatedly measured using the GC/TCD, and the $\delta\{(O_2+Ar)/N_2\}_{round\text{-}robin}$ values were calculated based on the ratios of the $(O_2 + Ar)$ peak area to $N_2$ peak area using Eq. (9).

$$\delta\{(O_2 + Ar)/N_2\}_{round\text{-}robin} = \frac{\{(O_2+Ar)/N_2\}_{round\text{-}robin}}{\{(O_2+Ar)/N_2\}_{ref}} - 1. \tag{9}$$

The $\delta(O_2/N_2)_{round\text{-}robin}$ value is given by Eq. (10).

$$\delta(O_2/N_2)_{round\text{-}robin} = (1 + a) \times \delta\{(O_2 + Ar)/N_2\}_{round\text{-}robin} - a \times \delta(Ar/N_2)_{round\text{-}robin}, \tag{10}$$

where the coefficient $a$ is defined by $a = k(Ar/O_2)_{ref}$. $k$ represents the TCD sensitivity ratio of Ar relative to $O_2$, and the value was evaluated as 1.13 by comparing gravimetric mixtures of $O_2 + N_2$ and $Ar + O_2 + N_2$ (Tohjima et al., 2005). Natural air is used for the reference gas. Therefore, the value of $a$ is calculated as 0.050 (Ar = 0.93% and $O_2$ = 20.94%). In this study, the

$\delta(Ar/N_2)_{round\text{-}robin}$ value was calculated based on $N_2$ molar fractions in the round-robin standard mixtures calculated based on $\delta\{(O_2+Ar)/N_2\}_{round\text{-}robin}$ values from the GC/TCD and $CO_2$ molar fractions from non-dispersive infrared spectroscopy and gravimetric Ar molar fractions in the round-robin standard mixtures.





The NIES $O_2/N_2$ scale is related to a set of 11 primary reference air. The NIES $O_2/N_2$ scale's long-term stability has been maintained within $\pm 0.45$ per meg yr$^{-1}$ by analysing the relative differences in the $O_2/N_2$ ratios in the primary and working reference air (Tohjima et al., 2019). Details of the analytical methods and the NIES $O_2/N_2$ scale are given in Tohjima et al. (2005, 2008).

**2.3.3 TU**

The $\delta(O_2/N_2)_{\text{round-robin}}$ values for TU were calculated based on the $\delta(^{16}O^{16}O/^{15}N^{14}N)_{\text{round-robin}}$ values measured using a mass spectrometer (Finnigan MAT-252). The $\delta(^{16}O^{16}O/^{15}N^{14}N)_{\text{round-robin}}$ values were calculated against the reference air on the TU scale, which is natural air filled in a 47 L manganese steel cylinder in 1998. The measurement technique's detail was given by Ishidoya et al. (2003). The mass spectrometer was adjusted to measure ion beam currents for masses 28 ($^{14}N^{14}N$), 29 ($^{15}N^{14}N$), 10 32 ($^{16}O^{16}O$), and 34 ($^{18}O^{16}O$). The $\delta(O_2/N_2)_{\text{NMIJ/AIST}}$ values are unequal to the isotopic ratios of $\delta(^{16}O^{16}O/^{15}N^{14}N)_{\text{round-robin}}$ measured by TU. Therefore, the $\delta(O_2/N_2)_{\text{round-robin}}$ values were calculated using the isotopic ratios $^{14}N^{14}N/^{15}N^{14}N$, $^{17}O^{16}O/^{16}O^{16}O$, and $^{18}O^{16}O/^{16}O^{16}O$, as shown in Eq. (11).

$$\delta(O_2/N_2)_{\text{round-robin}} = \left[\delta(^{16}O^{16}O/\,^{15}N\,^{14}N) + 1\right]_{\text{round-robin}} \times$$

$$\left[\frac{1+^{17}O^{16}O/^{16}O^{16}O+^{18}O^{16}O/^{16}O^{16}O}{1+^{14}N\,^{14}N/^{15}N^{14}N}\right]_{\text{round-robin}} \Big/ \left[\frac{1+^{17}O^{16}O/^{16}O^{16}O+^{18}O^{16}O/^{16}O^{16}O}{1+^{14}N\,^{14}N/^{15}N^{14}N}\right]_{\text{ref}} - 1 \qquad (11)$$

The isotopic ratios in the round-robin standard mixtures were calculated using Eqs. (6), (7), and (12).

$$^{14}N^{14}N/^{15}N^{14}N = \left[\delta(^{14}N^{14}N/^{15}N^{14}N)_{\text{round-robin}} + 1\right] \times (^{14}N^{14}N/^{15}N^{14}N)_{\text{ref}}. \qquad (12)$$

In this study, we used the values of $\delta(^{18}O^{16}O/^{16}O^{16}O)_{\text{round-robin}}$, $\delta(^{17}O^{16}O/^{16}O^{16}O)_{\text{round-robin}}$, and $\delta(^{14}N^{14}N/^{15}N^{14}N)_{\text{round-robin}}$ measured by EMRI/AIST, rather than by TU, to reduce the uncertainties of the $\delta(O_2/N_2)_{\text{round-robin}}$ values associated with the isotope ratio measurements. The $(^{18}O^{16}O/^{16}O^{16}O)_{\text{ref}}$, $(^{17}O^{16}O/^{16}O^{16}O)_{\text{ref}}$, and $(^{15}N^{14}N/^{14}N^{14}N)_{\text{ref}}$ values were calculated based on the corresponding atmospheric values, similar to the EMRI/AIST values.

**2.3.4 SIO**

SIO reported the $\delta(O_2/N_2)$ values based on measurements using a two-wavelength interferometer (Keeling et al., 1998). The SIO $O_2/N_2$ reference ($\delta(O_2/N_2) = 0$) is based on a suite of 18 primary reference gases stored in high-pressure cylinders (aluminium or steel, volumes ranging from 29 to 47 L) filled with natural air (Keeling et al., 2007). Differences between the round-robin cylinders and the SIO reference were determined from




$$\delta(O_2/N_2)_{round-robin} = \frac{1}{S_{O_2} \cdot X_{O_2}(1-X_{O_2})} \cdot \delta\tilde{r} - I_{CO_2} \cdot \Delta CO_2 - I_{Ar/N_2} \cdot \delta(Ar/N_2) - other\ interferences$$

(13)

where $\delta\tilde{r}$ is the difference in refractivity ratio $\tilde{r}$ = r(2537.27 Å)/r(4359.57 Å) between the round-robin cylinder and the SIO reference, determined via interferometric comparisons with secondary reference gases linked to the primary suite. $S_{O_2}$ =0.03397 is a constant sensitivity factor, $X_{O_2}$ is the mole fraction of the SIO reference, $I_{CO_2}$ is a constant (1.0919 per meg/ppm), and $\Delta CO_2$ is the difference in $CO_2$ mole fraction from the SIO reference (363.29 µmol mol⁻¹). SIO data are routinely corrected for $CO_2$ interference. We apply additional corrections for $Ar/N_2$, Ne, He, Kr, Xe, $CH_4$, $N_2O$, and CO. The additional corrections are effectively constant (or small) in natural air. They can usually be neglected in comparisons of natural air samples. However, these corrections cannot be neglected in relating the SIO scale to an absolute $O_2/N_2$ reference based on the round-robin cylinders, which may differ in their $Ar/N_2$ ratios from natural air and which lack constituents other than $N_2$, $O_2$, Ar, and $CO_2$. These corrections require estimates of the molar $Ar/N_2$ ratio and other gases' abundances in typical background air. Notably, the primary reference gases are relevant in Eq. (13) as references for relative refractivity. Therefore, the exact $Ar/N_2$ ratio and abundances of other gases in the SIO reference are not directly relevant. For background air, the following values were adopted: $Ar/N_2$ = 0.0119543, $Ne/N_2$ = 2.328 × 10⁻⁵, $He/N_2$ = 6.71×10⁻⁶, $Kr/N_2$ = 1.46×10⁻⁶, $Xe/N_2$ = 1.11×10⁻⁷, $CH_4$ = 1.8 µmol mol⁻¹, $N_2O$ = 0.3 µmol mol⁻¹, CO = 0.1 µmol mol⁻¹. Here, $Ar/N_2$ is from Aoki et al. (2019), and the other (noble gas)/$N_2$ ratios are from Glueckhauf (1951). The sensitivity $S_{O_2}$ and interference factors (e.g., $I_{Ar/N_2}$ = −0.0124) in Eq. (13) are based on refractivity data for the pure gases and natural air (Keeling, 1988, Keeling et al., 1998) using Xe data from Kronjäger (1936) (also see Keeling et al., 2020). The quantity $\delta(Ar/N_2)$ was computed using the AIST gravimetric data, $\delta(Ar/N_2)$ = (($Ar/N_2)_{grav}$/0.0119543 −1).

The $Ar/N_2$ interference ($-I_{Ar/N_2} \cdot \delta(Ar/N_2)$) ranges from −55 to + 24 per meg, depending on the round-robin cylinder. The sum of the remaining interferences, other than for $CO_2$ (- *other interferences*), is effectively constant at −14.3 per meg. The largest individual contributions are from Ne (−32.8 per meg) and $CH_4$ (+11.9 per meg).

## 3 Results and Discussion

### 3.1 Stability of δ(O₂/N₂) During Inter-comparison

The $\delta(O_2/N_2)_{round-robin}$ values were measured four times using the mass spectrometer by EMRI/AIST to evaluate the stability of the $O_2/N_2$ ratios of the standard mixtures during the inter-comparison experiment. The initial $\delta(O_2/N_2)_{round-robin}$ values in the measurement of four times were used as the EMRI/AIST assigned values. The $\delta(O_2/N_2)_{round-robin}$ values were calculated against the EMRI/AIST scale. The EMRI/AIST scale's stability was evaluated by measuring the values of $\delta(O_2/N_2)$ in three working





reference air against the primary reference air from 2012 to 2020. The changing rates and their standard deviations of $\delta(O_2/N_2)$ in the respective cylinders were $0.27 \pm 0.15$ per meg $yr^{-1}$, $0.16 \pm 0.23$ per meg $yr^{-1}$, $-0.38 \pm 0.25$ per meg $yr^{-1}$, and $0.08 \pm 0.11$ per meg $yr^{-1}$ on average. Therefore, the working standards show no systematic trend in $\delta(O_2/N_2)$ regarding the primary reference air.

Figure 1 shows the temporal drifts of the $\delta(O_2/N_2)_{\text{round-robin}}$ values from the initial values determined by the mass spectrometer at EMRI/AIST. The first measurement was conducted immediately after preparing the round-robin standard mixtures: May 2017 for three cylinders (CPB16345, CPB16315, CPB16379) and July 2017 for the other cylinders (CPB28912, CPB16349). The temporal drifts analysed in March 2018 (before shipment) ranged from $-5.9$ to $5.5$ per meg. This range was within the expanded uncertainty (6.4 per meg) of measurement using the mass spectrometer of EMRI/AIST. Here the expanded

uncertainty (a coverage factor of 2) represents $\approx$ a 95% level of confidence. The temporal drifts analysed in March 2019 (after the cylinder's return from SIO) ranged from $-16.4$ per meg to $2.9$ per meg. This range was larger than the expanded uncertainty of measurement.

We also analysed the round-robin standard mixtures in March 2020 (a year after return) and found that the temporal drifts ranged from $-18.3$ per meg to $-5.6$ per meg. The $\delta(O_2/N_2)_{\text{round-robin}}$ values decreased slightly with time in all cylinders, especially for cylinder no. CPB16379. The average decreasing rate of the $\delta(O_2/N_2)_{\text{round-robin}}$ values in the cylinders, except for

CPB16379, was $-3.2 \pm 1.1$ per meg $yr^{-1}$. Meanwhile, that of the CPB16379 cylinder was $-6.7 \pm 2.1$ per meg $yr^{-1}$. The decreasing rates and standard deviations were calculated from least-square fitting. The decrease in the $\delta(O_2/N_2)_{\text{round-robin}}$ values during the inter-comparison experiment are thought to be caused by $O_2$ consumption by the oxidation of residual organic material, oxidation of the inner surface of the cylinders, and selective $O_2$ desorption on the inner surface of the cylinders rather

than the fractionation of $O_2$ and $N_2$ since of the escape of gas from the cylinder generally increases the $O_2/N_2$ in a cylinder (Langenfelds et al., 1999). We corrected the temporal drifts during the inter-comparison experiment by linearly interpolating the $\delta(O_2/N_2)_{\text{NMIJ/AIST}}$ value of the data analysed by individual laboratories using the temporal drifts measured before and after the analysis of individual laboratories. Following this, we compared the interpolated $\delta(O_2/N_2)_{\text{NMIJ/AIST}}$ value with the measured $\delta(O_2/N_2)_{\text{round-robin}}$ value.

We evaluated the NMIJ/AIST scale's reproducibility using nine high-precision standard mixtures prepared in different periods (from April 2017 to February 2020). Figure 2 shows the relations between the $\delta(O_2/N_2)_{\text{NMIJ/AIST}}$ values gravimetrically determined by NMIJ/AIST and the $\delta(O_2/N_2)$ values measured using the mass spectrometer at EMRI/AIST. The lines in Figure 2a represent the Deming least-square fit to the data, and Figure 2b shows residuals of $\delta(O_2/N_2)_{\text{NMIJ/AIST}}$ from the line. The error bar represents the expanded uncertainty of the $\delta(O_2/N_2)_{\text{NMIJ/AIST}}$ values. The high-precision standard mixtures prepared in April

and June 2017 were selected from the round-robin standard mixtures. All residuals were within the expanded uncertainties, which were less than 8 per meg, identified that the NMIJ/AIST scale could be reproduced any time by preparing high-precision standard mixtures. Results show that a long-term temporal drift of each laboratory's $\delta(O_2/N_2)$ scale, which is determined against a reference natural air in a high-pressure cylinder, can be evaluated by comparing the reference air with high-precision standard mixtures by NMIJ/AIST.





### 3.2 Inter-comparison Between Laboratory Scales and Its Span Sensitivities

Table 3 summarises the $\delta(O_2/N_2)_{round\text{-}robin}$ values measured by individual laboratories. Notably, $\delta(O_2/N_2)_{round\text{-}robin}$ shown in Table 3 are the corrected values for the deviations in $Ar/N_2$ ratios and isotopic ratios of $N_2$ and $O_2$ in the round-robin standard mixtures from the atmospheric values and determined against their scales, as described in Section 2.3.

Figure 3a plots the relations between the $\delta(O_2/N_2)_{NMIJ/AIST}$ and $\delta(O_2/N_2)_{round\text{-}robin}$ values of individual laboratories. The $\delta(O_2/N_2)_{NMIJ/AIST}$ values were interpolated to correct the temporal drifts of $\delta(O_2/N_2)$, as described in Section 3.1. The lines represent a Deming least-square fit to the plotted data for individual laboratories (Table 4). The slopes and their standard deviations for EMRI/AIST, TU, NIES, and SIO were $0.9983 \pm 0.0010$, $0.9983 \pm 0.0013$, $1.0329 \pm 0.0013$, and $1.0087 \pm 0.0010$, respectively. The deviations from 1 for the slopes of the lines represent the differences from the NMIJ/AIST scale's span

sensitivity, which ranged from −0.17% to 3.3%. The intercepts of the lines represent the differences between individual laboratory scales and NMIJ/AIST scale corresponding to $\delta(O_2/N_2)_{NMIJ/AIST} = 0$: $-581.0 \pm 2.2$, $-221.4 \pm 3.1$, $-243.0 \pm 3.0$, and $-50.7 \pm 2.4$ per meg for EMRI/AIST, TU, NIES, and SIO, respectively. The results reflect the difference in the filling years of the primary standard of individual laboratories. The numbers following the symbol ± represent the standard deviations. The differences in the intercepts between SIO and other laboratories were $-530.3 \pm 3.3$, $-170.7 \pm 3.9$, and $-192.4 \pm 3.9$ per

meg for EMRI/AIST, TU, and NIES, respectively. The differences of NIES and TU from SIO were consistent with those obtained from past inter-comparison experiments (the GOLLUM comparison, 2015) (Table 4) although the difference of TU from SIO was slightly bigger. Figure 3b shows the residuals from the fitting lines. All of them fall within expanded uncertainties on the measurement for individual laboratories.

### 3.3 Compatibility of the Atmospheric $\delta(O_2/N_2)$ Data Between the Laboratories and Its Implication to the Global $CO_2$
Budget Analysis

This study shows that the inter-comparison results allow us to compare the observation data of individual laboratories directly. We compared the $O_2/N_2$ ratios measured by EMRI/AIST and NIES based on flask samples collected at HAT from October 2015 to December 2019 (Tohjima et al., 2008). The values of NIES after March 2018 are preliminary data. The air samples were collected twice monthly into two Pyrex glass flasks arranged in series (one for AIST and the other for NIES). We

confirmed that the isotopic ratios of $N_2$ and $O_2$ did not significantly differ from the atmospheric values for the HAT air samples. Therefore, we regard the values of $\delta(^{16}O^{16}O/^{14}N^{14}N)$ and $\delta\{(O_2+Ar)/N_2\}$ which were measured using the mass spectrometer and GC/TCD equal to $\delta(O_2/N_2)$ in Eq. (1). Figure 4a shows the $\delta(O_2/N_2)$ values reported on the NIES and EMRI/AIST scales. The average difference in the $\delta(O_2/N_2)$ between the two scales was $-329.3 \pm 6.9$ per meg. The uncertainty represents the standard deviation of the differences. Both values of $\delta(O_2/N_2)$ were converted to the NMIJ/AIST scale using Eq. (14),


$$\delta(O_2/N_2)_{NMIJ/AIST} = a_n \cdot \delta(O_2/N_2)_n + b_n, \tag{14}$$





where $a_n$ and $b_n$ are the slope and intercept of each laboratory's line ($n$) obtained in Section 3.2. Figure 4b shows the converted $\delta(O_2/N_2)$ values. This scale conversion reduced the bias between the $\delta(O_2/N_2)$ values of EMRI/AIST and NIES to $-6.6 \pm 6.8$ per meg (subtracting the $\delta(O_2/N_2)$ values of EMRI/AIST from those of NIES. The bias dropped within the uncertainty,

representing the standard deviation of the differences. Figures 5a and 5b plot both values of $\delta(O_2/N_2)$ before and after the scale conversion, confirming the compatibility between the span sensitivities on the EMRI/AIST and NIES scales. The lines represent a Deming least-square fit to the scatter plots. The slope of the line before scale conversion and its standard deviation is $0.956 \pm 0.015$, consistent with the difference in the span sensitivity between both scales ($0.9983/1.0329 = 0.967$) within uncertainty. After the scale conversion, the slope and its standard deviation is $0.990 \pm 0.015$, identifying that the scale

conversion corrected the difference in the span sensitivity between the EMRI/AIST and NIES scales to the NMIJ/AIST scales. Observing the long-term trend in atmospheric $\delta(O_2/N_2)$ provides critical information on the global $CO_2$ budget (Manning and Keeling, 2006). Recently, Tohjima et al. (2019) estimated the land biospheric and oceanic $CO_2$ uptakes using the average secular changing rate of $\delta(O_2/N_2)$ reported on the NIES scale. We converted the changing rate of $\delta(O_2/N_2)$ on the NIES scale to that on the NMIJ/AIST scales and recalculated the global $CO_2$ budgets from 2000 to 2016 using the converted rates. Table

5 summarises the $CO_2$ budgets reported by Tohjima et al. (2019) and recalculated by this study. Notably, the fossil fuel-derived $CO_2$ emissions and the global average of the atmospheric $CO_2$ molar fractions used for the $CO_2$ budget calculation are the same as those used in the Global Carbon Project for estimating the global carbon budget in 2018 (Le Quéré et al., 2018).
0.29 Pg yr$^{-1}$ corrected the land biospheric and oceanic $CO_2$ uptakes due to the scale conversions. These amounts correspond to 29% and 11% of the land biospheric and oceanic carbon budgets estimated by NIES and not negligible. Results show that

the span sensitivities of the $O_2/N_2$ scale are critical accurately estimating carbon budgets. Moreover, Resplandy et al. (2019) estimated an increase in the global OHC based on the atmospheric $O_2$ and $CO_2$ measurements. They reported that the largest single source of uncertainty in their estimation is the scale error from the span calibration of the $O_2/N_2$ analyser. They also mentioned that the error would be reduced via within-lab and inter-lab comparisons. Therefore, the span sensitivities of the EMRI/AIST, TU, NIES, and SIO scales against the NMIJ/AIST absolute scale obtained from the inter-comparison experiment

in this study should improve the accuracy of the OHC increase estimate significantly.

## 4 Conclusions

The inter-comparison experiment was used to evaluate the relationship between the measured $\delta(O_2/N_2)$ values and span sensitivities of the individual laboratory scales from the NMIJ/AIST scale using gravimetrically prepared high-precision standard mixtures. The deviations of the span sensitivities ranged from $-0.17\%$ to $3.3\%$, which were quantified for the first

time in the world. The difference between individual laboratory scales corresponds to the land biospheric and oceanic $CO_2$ uptake of 0.29 Pg yr$^{-1}$, which are not negligible. The deviations in the measured $\delta(O_2/N_2)$ values on the EMRI/AIST, TU, NIES, and SIO scales from the NMIJ/AIST scale corresponding to $\delta(O_2/N_2)_{NMIJ/AIST} = 0$ were $-581.0 \pm 2.2$, $-221.4 \pm 3.1$,





−243.0 ± 3.0, and −50.7 ± 2.4 per meg, respectively. The differences between individual absolute values were consistent with the results from the GOLLUM round-robin cylinder comparison. However, the $\delta(O_2/N_2)$ values in the round-robin standard mixtures decreased at rates of −6.7 ± 2.1 per meg yr$^{-1}$ for one cylinder and −3.2 ± 1.1 per meg yr$^{-1}$ for the other four cylinders. The decrease was caused by $O_2$ consumption by oxidation of residual organic material, oxidation of the cylinders' inner surface, and selective $O_2$ desorption on the inner surface of the cylinders rather. The fractionation of $O_2$ and $N_2$ did not cause it because of the escape of gas from the cylinder. The $O_2/N_2$ ratios in high-precision standard mixtures prepared in different periods by NMIJ/AIST are reproduced within the $O_2/N_2$ ratios' uncertainty, identifying that the NMIJ/AIST scale can be reproduced any time by preparing high-precision standard mixtures. Further, a long-term temporal drift of each laboratory's scale can be evaluated by comparing the reference air with high-precision standard mixtures prepared by NMIJ/AIST. Finally, we demonstrated that variations in the atmospheric $\delta(O_2/N_2)$ on the EMRI/AIST and NIES scales in flask samples collected at HAT became comparable by converting both scales to the NMIJ/AIST scale, although the bias is not negligible. The results obtained in this study should improve the estimation method of carbon budgets and OHC increase.

**Acknowledgments**

We thank the Global Environmental Forum (GEF) staff for their work in collecting the air samples at the Hateruma station. The Global Environment Research Account partly supported this study for the National Institute of the Ministry of the Environment, Japan and the JSPS KAKENHI Grant Number 19K05554.



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





**Table 1.** The gravimetric values of $N_2$, $O_2$, Ar, and $CO_2$ molar fractions and $\delta(O_2/N_2)$ in five round-robin standard mixtures prepared by the NMIJ/AIST[a]

[a] The high-precision

| Cylinder number | Preparation date | Gravimetric values[b] | | | | |
|---|---|---|---|---|---|---|
| | | $N_2$[c] | $O_2$[c] | Ar[c] | $CO_2$[c] | $\delta(O_2/N_2)$[d] |
| CPB16345 | April 7, 2017 | 781499.1 ± 1.0 | 208750.7 ± 0.8 | 9349.6 ± 0.7 | 400.43 ± 0.03 | −4226.7 ± 4.0 |
| CPB16315 | April 12, 2017 | 781264.6 ± 0.9 | 209040.2 ± 0.7 | 9297.0 ± 0.7 | 398.18 ± 0.03 | −2546.6 ± 3.8 |
| CPB16379 | April 17, 2017 | 781059.4 ± 0.8 | 209233.2 ± 0.7 | 9308.6 ± 0.6 | 398.68 ± 0.03 | −1363.2 ± 3.3 |
| CPB28912 | June 15, 2017 | 780792.2 ± 0.8 | 209437.1 ± 0.7 | 9351.1 ± 0.6 | 419.44 ± 0.03 | −47.9 ± 3.4 |
| CPB16349 | June 13, 2017 | 780424.6 ± 0.8 | 209813.5 ± 0.7 | 9342.7 ± 0.6 | 419.06 ± 0.03 | 2221.1 ± 3.4 |

standard mixtures were prepared in a previous study (Aoki et al., 2019). However, the gravimetric values of $N_2$, $O_2$,

Ar, and $CO_2$ molar fractions were recalculated based on the cylinders' expansion rate. The value was determined as 1.62 ±

0.06 ml $MPa^{-1}$ by our experiment (unpublished data).

[b] The numbers following the symbol ± denote the standard uncertainty.

[c] Figures are given in the unit of µmol $mol^{-1}$.

[d] Figures are given in the unit of per meg. These values were calculated against the $O_2/N_2$ ratio in the atmosphere

(0.20946/0.78084 = 0.26825) (Machta and Hughes, 1970).





**Table 2.** Measurement techniques, measurement species, and reported values of EMRI/AIST, NIES, TU, and SIO.

| Constituent | EMRI/AIST | NIES | TU | SIO |
|---|---|---|---|---|
| Analysis period | May–July 2017 | Sep–Nov 2017 | Dec 2017–Jan 2018 | May–Nov 2018 |
| Measurement technique | Mass spectrometry | Gas chromatography | Mass spectrometry | Interferometric method |
| Measurement species | $^{14}N^{14}N, ^{15}N^{14}N,$ $^{16}O^{16}O,$ $^{17}O^{16}O, ^{18}O^{16}O$ | $O_2, N_2, Ar$ | $^{16}O^{16}O, ^{14}N^{15}N$ | $O_2$ (interferometer) $^{40}Ar, ^{14}N^{14}N$ (mass spectrometer) |
| Reported values | $\delta(^{16}O^{16}O /^{14}N^{14}N)$[a] | $\delta(O_2/N_2)$ | $\delta(^{16}O^{16}O /^{15}N^{14}N)$[a] | $\delta(O_2/N_2)$ |

[a] The $\delta(O_2/N_2)$ values of EMRI/AIST and TU were computed using $\delta(^{17}O/^{16}O)$, $\delta(^{18}O/^{16}O)$, and $\delta(^{15}N/^{14}N)$ measured by EMRI/AIST (see text). $CO_2$ molar fractions measured by EMRI/AIST were used to correct $\delta(^{16}O^{16}O /^{15}N^{14}N)$ values.



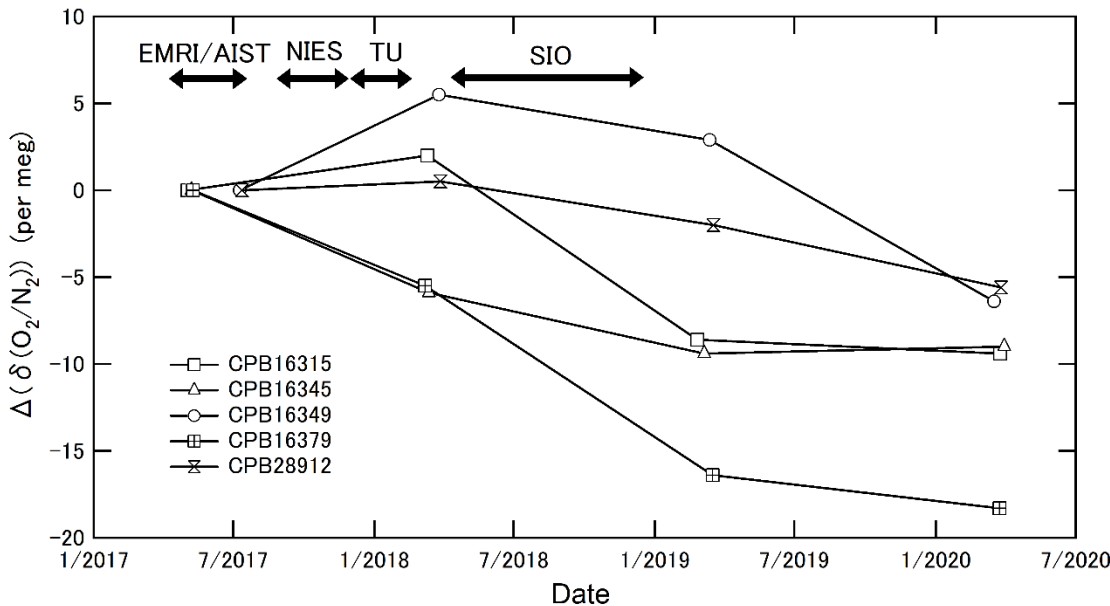

**Figure 1** The temporal drift of $\delta(O_2/N_2)_{\text{round-robin}}$ values from the initial values were measured using a mass spectrometer at
EMRI/AIST after preparing the round-robin standard mixtures before the shipment of the cylinders to SIO, after the return of
the cylinders from SIO, and a year after the return.





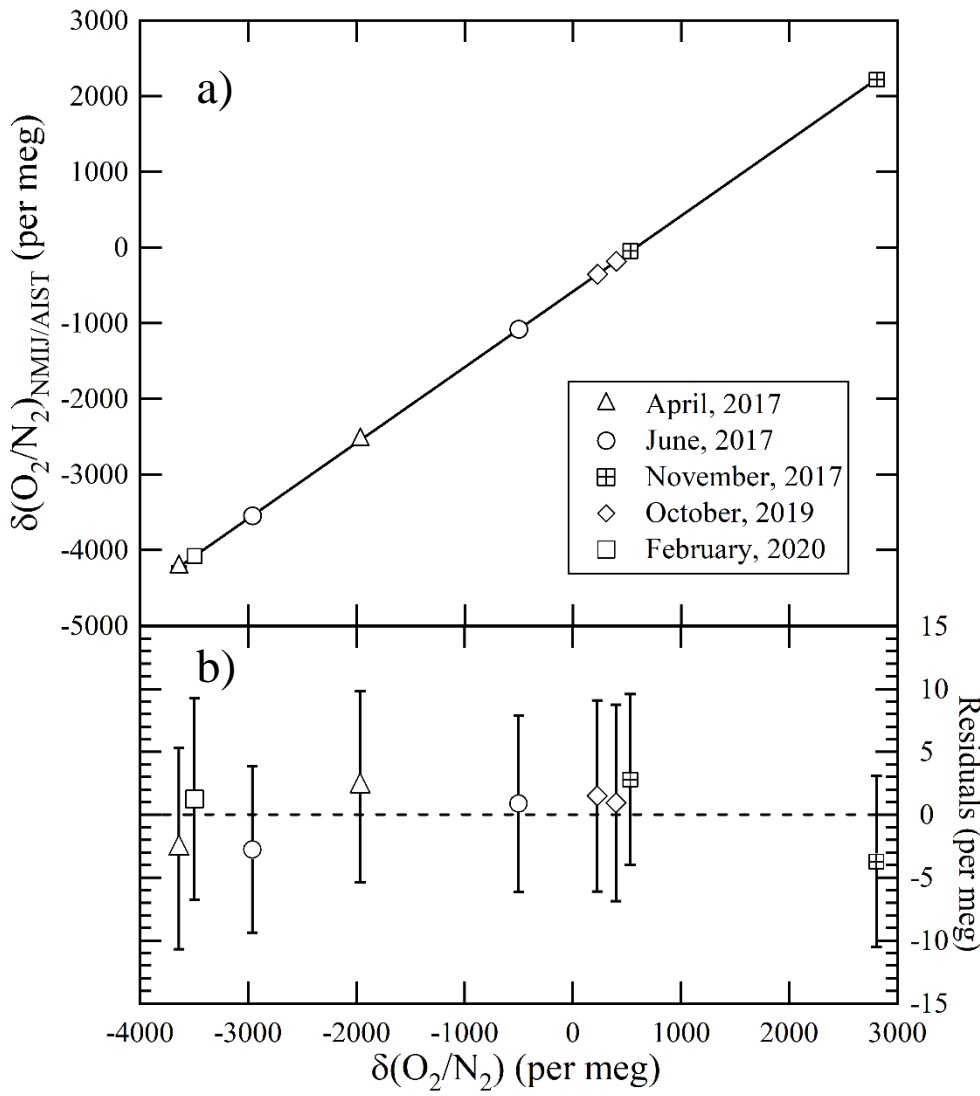

**Figure 2** a) Relationships between the $\delta(O_2/N_2)_{NMIJ/AIST}$ values of nine high-precision standard mixtures prepared from April 2017 to February 2020 and the $\delta(O_2/N_2)$ values measured using the mass spectrometer. b) Residuals from the line of the Deming least-square fit to the plots.





**Table 3** $\delta(O_2/N_2)_{round-robin}$ values in the round-robin standard mixtures reported by EMRI/AIST, NIES, TU, and SIO.

| Cylinder number | EMRI/AIST | NIES | TU | SIO |
|---|---|---|---|---|
| CPB16345 | −3647.7 ± 3.2 | −3859.4 ± 5.0 | −4014.6 ± 5.4 | −4141.7 ± 3.3 |
| CPB16315 | −1970.2 ± 3.2 | −2227.2 ± 5.0 | −2331.2 ± 5.4 | −2485.7 ± 3.3 |
| CPB16379 | −786.6 ± 3.2 | −1086.1 ± 5.0 | −1149.4 ± 5.4 | −1313.4 ± 3.3 |
| CPB28912 | 531.5 ± 3.2 | 183.1 ± 5.0 | 177.9 ± 5.4 | −0.4 ± 3.3 |
| CPB16349 | 2810.2 ± 3.2 | 2390.5 ± 5.0 | 2449.5 ± 5.4 | 2253.5 ± 3.3 |

Numbers are given in the unit of per meg. The numbers following the symbol ± denote the standard uncertainty of measurement for individual laboratories.




**Figure 3** a) Relationships between the $\delta(O_2/N_2)_{NMIJ/AIST}$ and $\delta(O_2/N_2)_{round\text{-}robin}$ values of EMRI/AIST, NIES, TU, and SIO and

lines obtained from the Deming least-square fit to the plotted data. b) Residuals of the $\delta(O_2/N_2)_{round\text{-}robin}$ values from the lines.



**Table 4.** Slopes and intercepts of the lines obtained by the Deming least-square fit to the reported $\delta(O_2/N_2)_{round\text{-}robin}$ values for individual laboratories, and deviation in the individual scales from SIO in this study and the GOLLUM 15.

| Institutes | Slopes $(a_n)$[a] | Intercepts $(b_n)$[b,c] | Deviation in individual scale from SIO scale[c,d] | Deviation from SIO values in the GOLLUM 15[c,e] |
|---|---|---|---|---|
| EMRI/AIST | $0.9983 \pm 0.0010$ | $-581.0 \pm 2.2$ | $-530.3 \pm 3.3$ | — |
| TU | $0.9983 \pm 0.0013$ | $-221.4 \pm 3.1$ | $-170.7 \pm 3.9$ | $-160 \pm 10.8$ |
| NIES | $1.0329 \pm 0.0013$ | $-243.0 \pm 3.0$ | $-192.4 \pm 3.9$ | $-195 \pm 10$ |
| SIO | $1.0087 \pm 0.0010$ | $-50.7 \pm 2.4$ | – | 0 |

Numbers following the symbol $\pm$ denote the standard uncertainty.

[a] Slope represents the difference in span sensitivity between individual laboratory scales and the NMIJ/AIST scale.

[b] Intercept represents a deviation in individual laboratory scale from the NMIJ/AIST scale corresponding to $\delta(O_2/N_2)_{NMIJ/AIST}$ = 0.

[c] Figures are given in the unit of per meg.

[d] Standard uncertainties were calculated by combining standard uncertainties of intercepts.

[e] Figures were summarised in the GOLLUM 15. EMRI/AIST did not participate in the GOLLUM 15.



**Figure 4** a) The δ(O₂/N₂) values obtained from the air samples collected at Hateruma Island for four years (2015–2019) measured by EMRI/AIST and NIES. b) The δ(O₂/N₂) values at Hateruma converted from EMRI/AIST and NIES scales to the NMIJ/AIST scale.





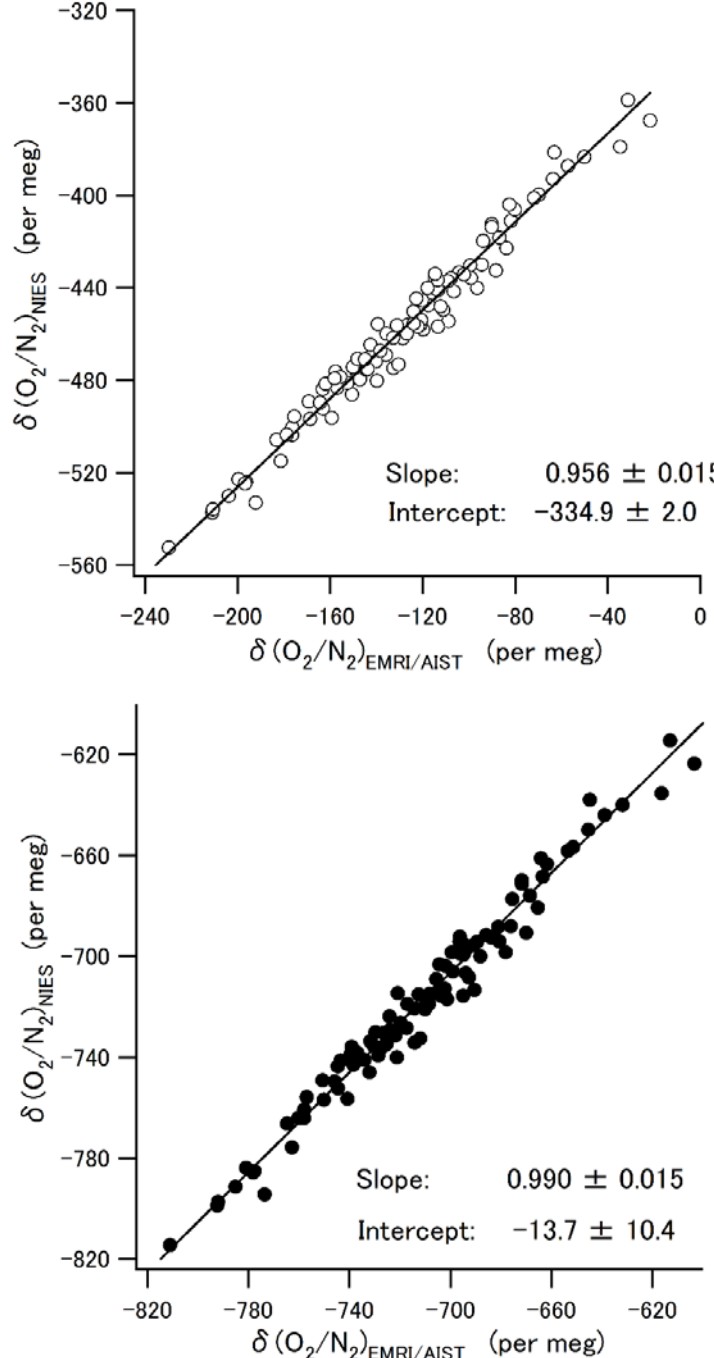

**Figure 5** a) Scatter plots of the δ(O$_2$/N$_2$) values at Hateruma for four years (2015–2019) on the EMRI/AIST and NIES scales.

The line represents the Deming least-square fit to the plots. b) Scatter plots between the δ(O$_2$/N$_2$) values for EMRI/AIST and NIES after conversion to the NMIJ/AIST scale. The line represents the Deming least-square fit to the plots.





**Table 5.** Land biospheric and oceanic $CO_2$ uptakes from 2000 to 2016 reported by Tohjima et al. (2019) on the NMIJ/AIST and NIES $O_2/N_2$ scales (see text for more details).

| | Fossil fuel[a] | Atm. $CO_2$[a] | Land uptake [b] | Ocean uptake [b] |
|---|---|---|---|---|
| NMIJ/AIST scale | | | 1.19 | 2.84 |
| | 8.48 | 4.45 | | |
| NIES scale | | | 1.48 (0.91) | 2.55 (0.73) |

Figures are given in units of PgC yr$^{-1}$

[a] These figures were from the Global Carbon Project (Le Quéré et al., 2018).

[b] NIES values were computed based on the average secular changing rate reported on the NIES scale by Tohjima et al. (2019). The figures in parentheses represent the uncertainties. NMIJ/AIST values were recalculated by converting the changing rate of $\delta(O_2/N_2)$ on the NIES scale to NMIJ/AIST scales.
