# Peer review of "Intercomparison of O2/N2 Ratio Scales Among AIST, NIES, TU, and SIO Based on Round-robin Exercise Using Gravimetric Standard Mixtures"

_Atmospheric Measurement Techniques, 2020_

## Referee Comment (RC2)

**Inter-comparison of O2/N2 Ratio Scales Among AIST, NIES, TU, and SIO Based on Round-Robin Using Gravimetric Standard Mixtures**

Nobuyuki Aoki1, Shigeyuki Ishidoya2, Yasunori Tohjima3, Shinji Morimoto4, Ralph F. Keeling5, Adam Cox5, Shuichiro Takebayashi4 and Shohei Murayama2

1National Metrology Institute of Japan (NMIJ), National Institute of Advanced Industrial Science and Technology (AIST), 1-1-1 Umezono, Tsukuba 305-8563, Japan 2Research Institute for Environmental Management Technology (EMRI), National Institute of Advanced Industrial Science

and Technology (AIST), Tsukuba 305-8569, Japan

3Center for Environmental Measurement and Analysis, National Institute for Environmental Studies, Tsukuba 305-8506, Japan

4Center for Atmospheric and Oceanic Studies, Graduate School of Science, Tohoku University, Sendai 980-8578, Japan 5Scripps Institution of Oceanography, La Jolla, CA 92093-0244, USA

Correspondence to: Nobuyuki Aoki (aoki-nobu@aist.go.jp)

Abstract. A study was conducted to compare the  $\delta(O_2/N_2)$  scales used by four laboratories engaged in atmospheric  $\delta(O_2/N_2)$

- 15 measurements. These laboratories are the Research Institute for Environmental Management Technology, Advanced Industrial Science and Technology (EMRI/AIST), the National Institute for Environmental Studies (NIES), Tohoku University (TU), and Scripps Institution of Oceanography (SIO). Therefore, five high-precision standard mixtures for O2 molar fraction gravimetrically prepared by the National Metrology Institute of Japan (NMIJ), AIST (NMIJ/AIST) with a standard uncertainty of less than 5 per meg were used as round-robin standard mixtures. EMRI/AIST, NIES, TU, and SIO reported the analysed
- 20 values of the standard mixtures on their own  $\delta(O_2/N_2)$  scales, and the values were compared with the  $\delta(O_2/N_2)$  values gravimetrically determined by NMIJ/AIST (the NMIJ/AIST scale). The  $\delta(O_2/N_2)$  temporal drift in the five standard mixtures during the inter-comparison experiment was corrected based on the  $\delta(O_2/N_2)$  values analysed before and after the experiments by EMRI/AIST. The scales are compared based on offsets in zero and span. The span offsets from the NMIJ/AIST scale ranged from -0.17% to 3.3%, corresponding with the difference of 0.29 Pg yr-1 in the standard biospheric and oceanic CO2
- 25

10

uptake5. The zero offsets from the NMIJ/AIST scale are  $-581.0 \pm 2.2$ ,  $-221.4 \pm 3.1$ ,  $-243.0 \pm 3.0$ , and  $-50.7 \pm 2.4$  per meg for EMRI/AIST, TU, NIES, and SIO, respectively. The atmospheric  $\delta(O_2/N_2)$  values observed at Hateruma Island (HAT; 24.05°N, 123.81°E), Japan, by EMRI/AIST and NIES became comparable by converting their scales to the NMIJ/AIST scale.

**Summary of Comments on amt-2020-481\_review.pdf**

**Page: 1**

| TNumber: 1                                      | Author: stephens     | Subject: Highlight Date: 5/21/2021 1:04:08 PM     |  |  |
|-------------------------------------------------|----------------------|---------------------------------------------------|--|--|
| intercomparison is                              | a word on its own (  | no hyphen needed)                                 |  |  |
| ·                                               | · · · ·              |                                                   |  |  |
| TNumber: 2                                      | Author: stephens     | Subject: Highlight Date: 5/21/2021 1:05:07 PM     |  |  |
| on a Round-Robir                                | n Exercise" might be | more informative / grammatically correct          |  |  |
|                                                 |                      |                                                   |  |  |
| 耳 Number: 3                                     | Author: stephens     | Subject: Cross-Out Date: 5/21/2021 1:08:53 PM     |  |  |
| a                                        | •                    | ·                                                 |  |  |
| u                                               |                      |                                                   |  |  |
| Number: 4                                       | Author: stephens     | Subject: Inserted Text Date: 5/21/2021 1:09:18 PM |  |  |
| global                                          | I                    |                                                   |  |  |
| 9.0.24                                          |                      |                                                   |  |  |
| The Number: 5                                   | Author: stephens     | Subject: Cross-Out Date: 5/21/2021 1:09:42 PM     |  |  |
| hased on trends in atmospheric (O2 and d(O2/N2) |                      |                                                   |  |  |
| based on denos in demospheric cor and d(or/ne)  |                      |                                                   |  |  |

**1. Introduction**

Observing the long-term change in atmospheric O2 molar fraction, combined with CO2 observation, enables us to estimate terrestrial biospheric and oceanic CO2 uptakes separately. O2 is exchanged with CO2 with some point point in terrestrial biospheric activities and fossil fuel combustion. Meanwhile, the ocean CO2 uptake and O2 emissions are decoupled since the ocean acts as a carbon sink by physiochemically dissolving the CO2 (e.g., Keeling et al., 1993). Various laboratories have performed ehanges in prospheric O2 since the early 1990s (e.g., Keeling et al., 1996; Bender et al., 2005; Manning and Keeling, 2006; Tohjima et al., 2008, 2019; Ishidoya et al., 2012a, b; Goto et al., 2017). Recently, Resplandy et al. (2019) introduced a method to estimate the global ocean heat content (OHC) increase based on atmospheric O2 and CO2 measurements. They extracted solubility-driven components of the atmospheric potential oxygen (APO = O2 + 1.1 × CO2) (Stephens et al., 2005).

10

1998) by combining their observational results with climate and ocean models. The global OHC change is a fundamental measure of global warming. Indeed, the ocean uptakes more than 90% of the easth's excess energy and is aluated based on ocean temperature measurements using Argo float 5.g., Levitus et al., 2012). Thus, the atmospheric O2 measurements are linked to the global CO2 budget and OHC.

The approaches described above rely on precision classical classica

$$\delta(O_2/N_2) = \frac{[n(O_2)/n(N_2)]_{sam}}{[n(O_2)/n(N_2)]_{ref}} - 1$$
(1)

25

In the equation, *n* depicts the molar amount of each substance, and the subscripts sam and ref represent sample and reference air, respectively. The  $\delta(O_2/N_2)$  value multiplied by 106 is expressed in per meg units. The O2 molar fractions in air are 0.946% (Machta and Hughes, 1970). Therefore, adding 1 µmol of O2 to a mole of dry air will increase in  $\delta(O_2/N_2)$  by 4.8 per meg.

30

Each laboratory has typically employed its own  $O_2/N_2$  reference based on natural air compressed and stored in high-pressure cylinders. Each laboratory has also assumed responsibility for calibrating the relationship between the measured instrument response and the reported change per meg units (span sensitivity). Therefore, the reported trends in  $O_2/N_2$  are potentially biased by any long-term drift in the  $O_2/N_2$  ratio of the reference cylinders (zero drift) or errors in the essured span sensitivity to that the uncertainty below 5 per meg is required for the global CO2 budget analyses based on

| 🔁 Number: 1          | Author: stephens       | Subject: Inserted Text          | Date: 5/21/2021 3:37:44 PM                          |
|----------------------|------------------------|---------------------------------|-----------------------------------------------------|
| distinct             |                        |                                 |                                                     |
| Number: 2            | Author: stephens       | Subject: Inserted Text          | Date: 5/21/2021 3:38:24 PM                          |
| measurements of      |                        |                                 |                                                     |
| TNumber: 3           | Author: stephens       | Subject: Inserted Text          | Date: 5/21/2021 3:38:59 PM                          |
| E                    |                        |                                 |                                                     |
| 🕆 Number: 4          | Author: stephens       | Subject: Inserted Text          | Date: 5/21/2021 3:39:15 PM                          |
| as                   |                        |                                 |                                                     |
| T Number: 5          | Author: stephens       | Subject: Inserted Text          | Date: 5/21/2021 3:39:21 PM                          |
| s                    |                        |                                 |                                                     |
| Number: 6            | Author: stephens       | Subject: Inserted Text          | Date: 5/21/2021 3:39:38 PM                          |
| e                    |                        |                                 |                                                     |
| T Number: 7          | Author: stephens       | Subject: Inserted Text          | Date: 5/21/2021 3:40:17 PM                          |
| а                    |                        |                                 |                                                     |
| TNumber: 8           | Author: stephens       | Subject: Highlight Date: 5/2    | 1/2021 3:43:41 PM                                   |
| cite Aoki et al., AM | T 2019 instead / also  | )                               |                                                     |
| 🚜 Author: step       | hens Subject: Stic     | ky Note Date: 5/21/2            | 2021 4:13:14 PM                                     |
| given global         | trends, it would also  | be helpful to give a year (196  | 7-70?) corresponding to this mole fraction estimate |
| TNumber: 9           | Author: stephens       | Subject: Highlight Date: 5/2    | 1/2021 3:44:58 PM                                   |
| previous sentence    | says this is calibrate | d, so "calibrated" or "measured | i" would be better there than "assumed"             |
| TNumber: 10          | Author: stephens       | Subject: Inserted Text          | Date: 5/21/2021 3:44:24 PM                          |
| of                   |                        |                                 |                                                     |
| TNumber: 11          | Author: stephens       | Subject: Cross-Out Date: 5/2    | 1/2021 3:47:40 PM                                   |
| a chan stability     |                        |                                 |                                                     |

a span stability

 $\delta(O_2/N_2)$  observations [Table 2 in Keeling et al. (1993)]. Challenges in achieving this precision clude fractionations of  $O_2$  and  $N_2$  induced by pressure, temperature, and water vapour gradients (Keeling et al., 2007), adsorption/desorption of the constituents on the cylinder's inner surface (Leuenberger et al., 2015), and permeation/leakage of the constituents from/through the valve (Sturm et al., 2004; Keeling et al., 2007). Tohjima et al. (2005) developed high-precision  $O_2$  standard mixtures with

5

25

30

15 per meg uncertainty for  $\delta(O_2/N_2)$  to resolve these problems by preparing gravimetric standard mixtures of pure N2,  $O_2$  Ar, and CO2. Their study was significant, but the uncertainties remain larger than those recommended by Keeling et al. (1993), as mentioned above.

Recently, a technique was developed for preparing high-precision primary standard mixtures with standard uncertainties less than 5 per meg for δ(O2/N2) at the National Metrology Institute of Japan, National Institute of Advanced Industrial Science
and Technology (NMIJ/AIST) (Aoki et al., 2019). The high-precision standard mixtures allow us to evaluate scale zero and span offsets accurately and precisely. In this study, we conducted interfection experiments to compare span sensitivities among the O2/N2 scales of figsearch Institute for Environmental Management Technology, Advanced Industrial Science and Technology (EMRI/AIST), National Institute for Environmental Studies (NIES), Tohoku University (TU), and Scripps Institution of Oceanography (SIO) using the developed high-precision standard mixtures. Following this, a regression analysis is applied to the inter-comparison results to investigate the relationship between the individual laboratory O2/N2 scales. Results showed a slight but significant difference in the span sensitivities of the individual scales. Finally, we compare the atmospheric δ(O2/N2) values observed on the EMRI/AIST scale with those on the NIES scale for the air samples collected at Hateruma Island (HAT; 24°03'N, 123°49'E), Japan, using the relationship between the individual laboratory scales obtained in this study.

**2. Experimental Procedures**

**20 2.1 NMIJ/AIST Scale and Round-Robin Standard Mixtures**

In this study, five high-precision standard mixtures with standard uncertainties less than 5 per meg for  $\delta(O_2/N_2)$  were used as round-robin standard mixtures. The NMIJ/AIST previously mixed them gravimetrically following ISO 6142-1:2015 (Aoki et al., 2019)<del>, which ge</del>re contained in 10 L aluminium-alloy cylinders (Luxfer Gas Cylinders, UK) with diaphragm valve (G-55, Hamai Industries Limited, Japan). Table 1 shows the gravimetrically determined molar fractions for N2, O2, Ar, CO2, as well as  $\delta(O_2/N_2)$  in the round-robin mixtures. However, the gravimetric values of N2, O2, Ar, and CO2 molar fractions were recalculated based on the cylinders' updated expansion rate. The value was determined as  $1.62 \pm 0.06$  ml Mpa-1 (unpublished data), which was determined by measuring pansion volume of a cylinder with an increase of inner pressure of the cylinders sunk in water since the previous expansion rate ( $2.2 \pm 0.2$  ml Mpa-1) was provided by a cylinder supplier. The source gases used are pure CO2 (>99.998%, Nippon Ekitan Corp., Japan), pure Ar (99.9999%, G1-grade, Japan Fine Products, Japan), pure O2 (99.99995%, G1-grade, Japan Fine Products, Japan).

Impurities in the source gases were identified and quantified via several techniques, including gas chromatography (GC). GC

| Page: 3                                 |                                              |                                                             |                                                                                                                              |
|-----------------------------------------|----------------------------------------------|-------------------------------------------------------------|------------------------------------------------------------------------------------------------------------------------------|
| 🔁 Number: 1                             | Author: stephens                             | Subject: Inserted Text                                      | Date: 5/21/2021 3:47:57 PM                                                                                                   |
| stability                               |                                              |                                                             |                                                                                                                              |
| TNumber: 2                              | Author: stephens                             | Subject: Inserted Text                                      | Date: 5/21/2021 3:49:10 PM                                                                                                   |
| in absolute terms                       |                                              |                                                             |                                                                                                                              |
| Hauthor: ste                            | phens Subject: Stic                          | cky Note Date: 5/2                                          | 1/2021 3:50:03 PM                                                                                                            |
| a bit tricky t                          | to talk about absolut                        | e uncertainty in per meg (a                                 | relative unit) - perhaps give in mole fraction instead?                                                                      |
| TNumber: 3                              | Author: stephens                             | Subject: Inserted Text                                      | Date: 5/21/2021 3:48:58 PM                                                                                                   |
|                                         |                                              |                                                             |                                                                                                                              |
| Thumber: 4                              | Author: stephens                             | Subject: Cross-Out Date:                                    | 5/21/2021 3:53:16 PM                                                                                                         |
|                                         |                                              |                                                             |                                                                                                                              |
| Number: 5                               | Author: stephens                             | Subject: Highlight Date:                                    | 5/24/2021 2:31:19 PM                                                                                                         |
| Here, it would be
addressed. Just ch | good to point out (a:
ecking against them | s you do below) that by con
once can fix span biases, bu | nparing to gravimetric mixtures prepared over time, the trend uncertainty can be
ut not necessarily zero drift over time. |
| TNumber: 6                              | Author: stephens                             | Subject: Inserted Text                                      | Date: 5/21/2021 3:53:33 PM                                                                                                   |
| the                                     |                                              |                                                             |                                                                                                                              |
| TNumber: 7                              | Author: stephens                             | Subject: Highlight Date:                                    | 5/21/2021 4:03:19 PM                                                                                                         |
| give materials?                         |                                              |                                                             |                                                                                                                              |
| TNumber: 8                              | Author: stephens                             | Subject: Inserted Text                                      | Date: 5/21/2021 4:02:54 PM                                                                                                   |
| . They                                  |                                              |                                                             |                                                                                                                              |
| TNumber: 9                              | Author: stephens                             | Subject: Inserted Text                                      | Date: 5/21/2021 4:04:35 PM                                                                                                   |
| the                                     |                                              |                                                             |                                                                                                                              |

equipped with a thermal conductivity detector (GC/TCD) was used to analyse  $N_2$ ,  $O_2$ ,  $CH_4$ , and  $H_2$  in pure  $CO_2$ .  $O_2$  and Ar in pure  $N_2$  and  $N_2$  in pure  $O_2$  were analysed using GC, equipped with a mass spectrometer. A Fourier-transform infrared spectrometer was used to detect  $CO_2$ ,  $CH_4$ , and CO in pure  $N_2$ ,  $O_2$ , and Ar. A galvanic cell  $O_2$  analyser was used to quantify  $O_2$  in pure Ar. A capacitance-type moisture sensor measured  $H_2O$  in pure  $CO_2$ , and a cavity ring-down moisture analyser measured  $H_2O$  in pure  $N_2$ ,  $O_2$ , and Ar.

5

10

In this study, the absolute  $O_2/N_2$  scale determined using the round-robin standard mixtures is hereafter the NMIJ/AIST scale. The NMIJ/AIST scale is presented only for scientific research and is uncertified by NMIJ. Here,  $\delta(O_2/N_2)_{\text{NMIJ/AIST}}$  represents the  $\delta(O_2/N_2)$  on the NMIJ/AIST scale, which was calculated against a reference  $O_2/N_2$  ratio of 12.20946/0.78084 = 0.26825, previously reported (Machta and Hughes, 1970). The range of  $\delta(O_2/N_2)_{\text{NMIJ/AIST}}$  values for the round-robin standard mixtures was -4200 per meg to 2200 per meg. The standard uncertainties of the  $\delta(O_2/N_2)_{\text{NMIJ/AIST}}$  values were 3.3 per meg to 4.0 per meg.

**2.2 Procedure of Inter2 comparison**

The EMRI/AIST, NIES, TU, and SIO conducted the intel3 comparison experiment. Five round-robin standard mixtures were analysed in the order of EMRI/AIST (May to July 2017), NIES (September to November 2017), TU (December 2017 to January 2018), and SIO (May to December 2018). Each lab reported the δ(O2/N2)round-robin values were aftermined against their scales to the NMIJ/AIST. The subscript round-robin is hereafter the round-robin5 standard mixture. Each lab analysed air delivered from the cylinders after placing them horizontally for more than five days after their transport to avoid the fange of δ(O2/N2)round-robin values in the standard mixtures by thermal diffusion and gravitational fractionation. The δ(O2/N2)round-robin values determined by individual laboratories using their methods were compared with the δ(O2/N2)NMIJ/AIST values.
EMRI/AIST and TU used mass spectrometry, NIES used GC, and SIO used the interferometric method, as summarised in Table 2. The stability of O2/N2 ratios in the round-robin standard mixtures during the inter-comparison experiment was evaluated by analysing their δ(O2/N2)round-robin values using a mass spectrometer (Delta-V, Thermo Fisher Scientific Inc., USA) (Ishidoya and Murayama, 2014) at EMRI/AIST before and after the inter1 comparison experiment.

Ar molar fractions in the round-robin standard mixtures were from 9297 to 9351  $\mu$ mol mol-1, much more variable than 25 variations in the tropospheric air (less than 1  $\mu$ mol mol-1) (Keeling et al., 2004). Isotopic ratios of  $\delta(^{17}O/^{16}O)$ ,  $\delta(^{18}O/^{16}O)$ , and  $\delta(^{15}N/^{14}N)$  in the round-robin standard mixtures, measured using the mass spectrometer by EMRI/AIST, there lower than the atmospheric values by 4.7‰, 9‰, and 2.4‰, respectively.  $\delta(^{17}O/^{16}O)$ ,  $\delta(^{18}O/^{16}O)$ , and  $\delta(^{15}N/^{14}N)$  are expressed as

$$\delta({}^{17}0/{}^{16}0) = \frac{[n({}^{17}0)/n({}^{16}0)]_{sam}}{[n({}^{17}0)/n({}^{16}0)]_{ref}} - 1$$
(2)

$$\delta({}^{18}0/{}^{16}0) = \frac{\left[n({}^{18}0)/n({}^{16}0)\right]_{sam}}{\left[n({}^{18}0)/n({}^{16}0)\right]_{ref}} - 1$$
(3)

$$\delta({}^{15}\mathrm{N}/{}^{14}\mathrm{N}) = \frac{[n({}^{15}\mathrm{N})/n({}^{14}\mathrm{N})]_{\mathrm{sam}}}{[n({}^{15}\mathrm{N})/n({}^{14}\mathrm{N})]_{\mathrm{ref}}} - 1.$$
(4)

30

Number: 1 Author: stephens Subject: Highlight Date: 5/21/2021 4:15:30 PM
 Why not use a more modern O2 mole fraction estimate? If it does not matter, consider saying so. Since you have gravimetric determinations, I'm guessing the assignment of zero on the scale is arbitrary, so it might help to say "we arbitrarily assign zero on the NMIJ/AIST scale to correspond to a ratio of 0.26825" and also maybe that this corresponds to the late 1960s.
 Rumber: 2 Author: stephens Subject: Cross-Out Date: 5/21/2021 4:18:42 PM

| T          |                        |                                                     |
|-------------------|------------------------|-----------------------------------------------------|
| TNumber: 3        | Author: stephens       | Subject: Cross-Out Date: 5/21/2021 4:19:00 PM       |
|                   | Authors stophone       | Subjects Inserted Text Date: E (21/2021 4:10:29 DM  |
| as                | Author: stephens       | Subject. Inserted Text Date. 5/21/2021 4.19.20 Pivi |
| Number: 5         | Author: stephens       | Subject: Highlight Date: 5/24/2021 2:32:12 PM       |
| necessary or coul | ld it be removed throu | ighout?                                             |
| Number: 6         | Author: stephens       | Subject: Inserted Text Date: 5/21/2021 4:21:13 PM   |
| TNumber: 7        | Author: stephens       | Subject: Cross-Out Date: 5/21/2021 4:24:01 PM       |
| Number: 8         | Author: stephens       | Subject: Highlight Date: 5/24/2021 10:41:50 AM      |

considering giving values on established reference scales

Here, the isotopic ratios of  $\delta({}^{17}\text{O}/{}^{16}\text{O})$ ,  $\delta({}^{18}\text{O}/{}^{16}\text{O})$ , and  $\delta({}^{15}\text{N}/{}^{14}\text{N})$  were approximately equal to those of  $\delta({}^{17}\text{O}/{}^{16}\text{O}/{}^{16}\text{O})$ .  $\delta(^{18}O^{16}O^{16}O^{16}O)$ , and  $\delta(^{15}N^{14}N^{14}N^{14}N)$ . This is because  $^{17}O^{17}O^{16}O^{16}O$ ,  $^{18}O^{18}O^{16}O^{16}O$ , and  $^{15}N^{15}N^{14}N^{14}N$  tended to be lower than 17O16O/16O/16O, 18O16O/16O/16O, and 15N14N/14N14N by 5000 times, 1000 times, and 500 times, respectively.

- $\square$  applied the following corrections to the measured  $\delta(O_2/N_2)_{round-robin}$  values from the individual laboratories by considering 5 the deviations of Ar molar fraction and isotopic ratios in the round-robin standard mixtures from the tropospheric air. The  $\delta(O_2/N_2)_{round-robin}$  values reported by EMRI/AIST and TU were corrected based on the deviation in the isotope ratio from the atmospheric level using isotopic ratios of N and O measured simultaneously at EMRI/AIST. This is because they measured the values of  $\delta({}^{16}O){}^{14}O){}^{14}N)$  and  $\delta({}^{16}O){}^{16}O){}^{14}N{}^{15}N)$ , respectively. NIES corrected t7 Ar molar fraction difference from its atmospheric level since the O2 peak obtained in GC included the Ar peak. SIO also corrected t3 difference in the Ar molar 10 fraction using the round-robin standard mixtures' gravimetric values since they only measured O2 molar fractions. The

measurement techniques and calculation procedures of the  $\delta(O_2/N_2)_{round-robin}$  values for individual laboratories are detailed in the next section.

**2.3 Analytical and Calculation Methods of $\delta(O_2/N_2)$ Values**

**2.3.1 EMRI/AIST**

15 The  $\delta(O_2/N_2)_{round-robin}$  values for EMRI/AIST were calculated based on the  $\delta({}^{16}O^{16}O^{14}N)_{round-robin}$  values measured using the mass spectrometer. The  $\delta({}^{16}O){}^{14}N)_{round-robin}$  values were calculated against the reference air on the EMRI/AIST scale, which is natural air filled in a 48 L aluminium cylinder with a diaphragm valve (G-55, Hamai Industries Limited, Japan). The measurement technique's detail was given in Ishidoya and Murayama (2014). The mass spectrometer was adjusted to measure ion beam currents for masses 28 (14N14N), 29 (15N14N), 32 (16O16O), 33 (17O16O), 34 (18O16O), and 44 (12C16O16O). The 20  $\delta(O_2/N_2)_{\text{NMIJ/AIST}}$  in the round-robin standard mixtures comprising all isotopes of O2 and N2 are unequal to the isotopic ratios of  $\delta({}^{16}O{}^{16}O{}^{14}N{}^{14}N)_{round-robin}$  measured using the mass spectrometer. Thus, mass-spectrometry-based isotopic ratios must be converted to values equivalent to the  $\delta(O_2/N_2)_{\text{NMIJ/AIST}}$  values. The  $\delta(O_2/N_2)_{\text{round-robin}}$  values were 4alculated based on isotopic ratios 15N14N/14N, 17O16O/16O/16O, and 18O16O/16O16O in the round-robin standard mixtures and reference air, as shown in Eq. (5).

25

30

$$\delta(0_2/N_2)_{\text{round-robin}} = \left[ \delta(^{16}0^{16}0/^{14}N) + 1 \right]_{\text{round-robin}} \times \left[ \frac{1+^{17}0^{16}0/^{16}0^{16}0+^{18}0^{16}0/^{16}0^{16}0}{1+^{15}N} \right]_{\text{round-robin}} / \frac{\left[ \frac{1+^{17}0^{16}0/^{16}0+^{18}0^{16}0/^{16}0^{16}0}{1+^{15}N} \right]_{\text{ref}} - 1.$$
(5)

Here, isotopic species of 17O17O, 18O17O, 18O18O, and 15N15N were negligible since their abundance was sufficiently small. The isotopic ratios of  ${}^{15}N{}^{14}N{}^{14}N{}^{14}N{}^{17}O{}^{16}O{}^{16}O{}^{16}O{}$ , and  ${}^{18}O{}^{16}O{}^{16}O{}^{16}O{}$  in the round-robin standard mixtures were calculated using Eqs. (6), (7), and (8).

| TNumber: 1                                                                              | Author: stephens      | Subject: Highlight Date: 5  | /24/2021 10:46:20 AM              |  |
|-----------------------------------------------------------------------------------------|-----------------------|-----------------------------|-----------------------------------|--|
| might be helpful t                                                                      | o start by saying why | these corrections are neede | ed - dilution, peak overlap, etc. |  |
| Number: 2                                                                               | Author: stephens      | Subject: Inserted Text      | Date: 5/24/2021 2:32:36 PM        |  |
| for (first sentence says thesea are O2/N2 corrections, not Ar corrections)              |                       |                             |                                   |  |
| T Number: 3                                                                             | Author: stephens      | Subject: Inserted Text      | Date: 5/24/2021 2:32:42 PM        |  |
| for (first sentence says thesea are O2/N2 corrections, not Ar corrections)              |                       |                             |                                   |  |
| TNumber: 4                                                                              | Author: stephens      | Subject: Highlight Date: 5  | /24/2021 10:48:22 AM              |  |
| say 'corrected' here instead? (first sentence already says 'calculated based on' 16/14) |                       |                             |                                   |  |

$${}^{18}O^{16}O^{/16}O^{16}O = [\delta({}^{18}O^{16}O^{/16}O^{16}O)_{\text{round-robin}} + 1] \times ({}^{18}O^{16}O^{/16}O^{16}O)_{\text{ref}},$$
(6)

$${}^{17}O^{16}O^{16}O^{16}O = [\delta({}^{17}O^{16}O^{16}O)_{\text{round-robin}} + 1] \times ({}^{17}O^{16}O^{16}O)_{\text{ref}},$$
(7)

$${}^{15}N^{14}N^{14}N^{14}N = [\delta({}^{15}N^{14}N^{14}N^{14}N)_{\text{round-robin}} + 1] \times ({}^{15}N^{14}N^{14}N^{14}N)_{\text{ref.}}$$
(8)

The isotopic ratios of  $\delta(^{15}N^{14}N/^{14}N^{14}N)_{round-robin}$ ,  $\delta(^{17}O^{16}O/^{16}O^{16}O)_{round-robin}$ , and  $\delta(^{18}O^{16}O/^{16}O)_{round-robin}$  were determined against the EMRI/AIST reference air. Values of  $(^{18}O^{16}O/^{16}O^{16}O)_{ref}$ ,  $(^{17}O^{16}O/^{16}O^{16}O)_{ref}$ , and  $(^{15}N^{14}N/^{14}N^{14}N)_{ref}$  refer to ratios of  $^{18}O^{16}O/^{16}O^{16}O_{16}O_{16}O_{16}O_{16}O_{16}O_{16}O_{16}O_{16}O_{16}O_{16}O_{16}O_{16}O_{16}O_{16}O_{16}O_{16}O_{16}O_{16}O_{16}O_{16}O_{16}O_{16}O_{16}O_{16}O_{16}O_{16}O_{16}O_{16}O_{16}O_{16}O_{16}O_{16}O_{16}O_{16}O_{16}O_{16}O_{16}O_{16}O_{16}O_{16}O_{16}O_{16}O_{16}O_{16}O_{16}O_{16}O_{16}O_{16}O_{16}O_{16}O_{16}O_{16}O_{16}O_{16}O_{16}O_{16}O_{16}O_{16}O_{16}O_{16}O_{16}O_{16}O_{16}O_{16}O_{16}O_{16}O_{16}O_{16}O_{16}O_{16}O_{16}O_{16}O_{16}O_{16}O_{16}O_{16}O_{16}O_{16}O_{16}O_{16}O_{16}O_{16}O_{16}O_{16}O_{16}O_{16}O_{16}O_{16}O_{16}O_{16}O_{16}O_{16}O_{16}O_{16}O_{16}O_{16}O_{16}O_{16}O_{16}O_{16}O_{16}O_{16}O_{16}O_{16}O_{16}O_{16}O_{16}O_{16}O_{16}O_{16}O_{16}O_{16}O_{16}O_{16}O_{16}O_{16}O_{16}O_{16}O_{16}O_{16}O_{16}O_{16}O_{16}O_{16}O_{16}O_{16}O_{16}O_{16}O_{16}O_{16}O_{16}O_{16}O_{16}O_{16}O_{16}O_{16}O_{16}O_{16}O_{16}O_{16}O_{16}O_{16}O_{16}O_{16}O_{16}O_{16}O_{16}O_{16}O_{16}O_{16}O_{16}O_{16}O_{16}O_{16}O_{16}O_{16}O_{16}O_{16}O_{16}O_{16}O_{16}O_{16}O_{16}O_{16}O_{16}O_{16}O_{16}O_{16}O_{16}O_{16}O_{16}O_{16}O_{16}O_{16}O_{16}O_{16}O_{16}O_{16}O_{16}O_{16}O_{16}O_{16}O_{16}O_{16}O_{16}O_{16}O_{16}O_{16}O_{16}O_{16}O_{16}O_{16}O_{16}O_{16}O_{16}O_{16}O_{16}O_{16}O_{16}O_{16}O_{16}O_{16}O_{16}O_{16}O_{16}O_{16}O_{16}O_{16}O_{16}O_{16}O_{16}O_{16}O_{16}O_{16}O_{16}O_{16}O_{16}O_{16}O_{16}O_{16}O_{16}O_{16}O_{16}O_{16}O_{16}O_{16}O_{16}O_{16}O_{16}O_{16}O_{16}O_{16}O_{16}O_{16}O_{16}O_{16}O_{16}O_{16}O_{16}O_{16}O_{16}O_{16}O_{16}O_{16}O_{16}O_{16}O_{16}O_{16}O_{16}O_{16}O_{16}O_{16}O_{16}O_{16}O_{16}O_{16}O_{16}O_{16}O_{16}O_{16}O_{16}O_{16}O_{16}O_{16}O_{16}O_{16}O_{16}O_{16}O_{16}O_{16}O_{16}O_{16}O_{16}O_{16}O_{16}O_{16}O_{16}O_{1$

**2.3.2 NIES**

15 NIES reported the  $\delta(O_2/N_2)_{round-robin}$  values based on the  $\delta\{(O_2+Ar)/N_2\}_{round-robin}$  values measured using a GC/TCD (Tohjima, 2000). The  $\delta\{(O_2+Ar)/N_2\}_{round-robin}$  values were calculated against the reference air on the NIES scale, which is natural air filled in a 48 L aluminium cylinder. A column separates the  $(O_2 + Ar)$  and  $N_2$  in the air sample, and a TCD detected the individual peaks. The reference and sample air were repeatedly measured using the GC/TCD, and the  $\delta\{(O_2+Ar)/N_2\}_{round-robin}$  values were calculated based on the ratios of the  $(O_2 + Ar)$  peak area to  $N_2$  peak area using Eq. (9).

20

5

10

$$\delta\{(O_2 + Ar)/N_2\}_{\text{round-robin}} = \frac{\{(O_2 + Ar)/N_2\}_{\text{round-robin}}}{\{(O_2 + Ar)/N_2\}_{\text{ref}}} - 1.$$
(9)

The  $\delta(O_2/N_2)$  round-robin value is given by Eq. (10).

25
$$\delta(O_2/N_2)_{\text{round-robin}} = (1+a) \times \delta\{(O_2 + Ar)/N_2\}_{\text{round-robin}} - a \times \delta(Ar/N_2)_{\text{round-robin}}, \tag{10}$$

where the coefficient *a* is defined by  $a = k(Ar/O_2)_{ref.} k$  represents the TCD sensitivity ratio of Ar relative to O2, and the value was evaluated as 1.13 by comparing gravimetric mixtures of O2 + N2 and Ar + O2 + N2 (Tohjima et al., 2005). Natural air is used for the reference gas. Therefore, the value of *a* is calculated as 0.050 (Ar = 0.93% and O2 = 20.94%). An this study, the  $\delta(Ar/N_2)_{round-robin}$  value was calculated based on N2 molar fractions in the round-robin standard mixtures calculated based on

30  $\delta(Ar/N_2)_{round-robin}$  value was calculated based on N2 molar fractions in the round-robin standard mixtures calculated based on  $\delta\{(O_2+Ar)/N_2\}_{round-robin}$  values from the GC/TCD and CO2 molar fractions from non-dispersive infrared spectroscopy and gravimetric Ar molar fractions in the round-robin standard mixtures.

| TNumber: 1 | Author: stephens | Subject: Inserted Text | Date: 5/24/2021 10:50:50 AM |
|------------|------------------|------------------------|-----------------------------|
| nearly (?) |                  |                        |                             |

Number: 2 Author: stephens Subject: Highlight Date: 5/24/2021 10:55:03 AM consider rewording for clarity - if you already have gravimetric Ar and N2, why do you need (O2+Ar)/N2 and CO2 to get (Ar/N2)? Also, change "in this study" to "For NIES" or equivalent.

The NIES  $O_2/N_2$  scale is related to a set of 11 primary reference air the NIES  $O_2/N_2$  scale's long-term stability has been maintained within ±0.45 per meg yr-1 analysing the relative differences in the  $O_2/N_2$  ratios in the primary and working reference air (Tohjima et al., 2019). Details of the analytical methods and the NIES  $O_2/N_2$  scale are given in Tohjima et al. (2005, 2008).

**5 **2.3.3 TU**

The  $\delta(O_2/N_2)_{round-robin}$  values for TU were calculated based on the  $\delta({}^{16}O^{16}O^{\square}N^{14}N)_{round-robin}$  values measured using a mass spectrometer (Finnigan MAT-252). The  $\delta({}^{16}O^{16}O^{/15}N^{14}N)_{round-robin}$  values were calculated against the reference air on the TU scale, which is natural air filled in a 47 L manganese steel cylinder in 1998. The measurement technique's detail was given by Ishidoya et al. (2003). The mass spectrometer was adjusted to measure ion beam currents for masses 28 ( ${}^{14}N^{14}N$ ), 29 ( ${}^{15}N^{14}N$ ), 32 ( ${}^{16}O^{16}O$ ), and 34 ( ${}^{18}O^{16}O$ ). The  $\delta(O_2/N_2)_{NMLI/AIST}$  values are unequal to the isotopic ratios of  $\delta({}^{16}O^{16}O^{/15}N^{14}N)_{round-robin}$ measured by TU. Therefore, the  $\delta(O_2/N_2)_{round-robin}$  values were calculated using the isotopic ratios  ${}^{14}N^{14}N^{/15}N^{14}N$ ,  ${}^{17}O^{16}O^{/16}O^{16}O$ , and  ${}^{18}O^{16}O^{/16}O^{16}O$ , as shown in Eq. (11).

$$\delta(O_2/N_2)_{\text{round-robin}} = \left[ \delta(^{16}O^{16}O/^{15}N^{14}N) + 1 \right]_{\text{round-robin}} \times$$

$$\quad \left[ \frac{1 + {}^{17}O^{16}O/{}^{16}O^{16}O + {}^{18}O^{16}O/{}^{16}O^{16}O}}{1 + {}^{14}N^{14}N/{}^{15}N^{14}N} \right]_{\text{round-robin}} / \left[ \frac{1 + {}^{17}O^{16}O/{}^{16}O^{16}O + {}^{18}O^{16}O/{}^{16}O^{16}O}}{1 + {}^{14}N^{14}N/{}^{15}N^{14}N} \right]_{\text{ref}} - 1$$

$$(11)$$

The isotopic ratios in the round-robin standard mixtures were calculated using Eqs. (6), (7), and (12).

$${}^{14}N^{14}N^{15}N^{14}N = [\delta({}^{14}N^{14}N^{15}N^{14}N)_{\text{round-robin}} + 1] \times ({}^{14}N^{14}N^{15}N^{14}N)_{\text{ref.}}$$
(12)

20

10

In this study, we used the values of  $\delta({}^{18}O^{16}O/{}^{16}O^{16}O)_{round-robin}$ ,  $\delta({}^{17}O^{16}O/{}^{16}O)_{round-robin}$ , and  $\delta({}^{14}N^{14}N/{}^{15}N^{14}N)_{round-robin}$  measured by EMRI/AIST, rather than by TU, to reduce the uncertainties of the  $\delta(O_2/N_2)_{round-robin}$  values associated with the isotope ratio measurements. The  $({}^{18}O^{16}O/{}^{16}O)_{ref}$ ,  $({}^{17}O^{16}O/{}^{16}O)_{ref}$ , and  $({}^{15}N^{14}N/{}^{14}N)_{ref}$  values were calculated based on the corresponding atmospheric values, similar to the EMRI/AIST values.

**25 **2.3.4**510**

SIO reported the  $\delta(O_2/N_2)$  values based on measurements using a two-wavelength interferometer (Keeling et al., 1998). The SIO  $O_2/N_2$  reference ( $\delta(O_2/N_2) = 0$ ) is based on a suite of 18 primary reference gases stored in high-pressure cylinders (aluminium or steel, volumes ranging from 29 to 47 L) filled with natural air (Keeling et al., 2007). Differences between the round-robin cylinders and the SIO reference were determined from

Table 2 mentions mass-spec measurements of 40Ar and 14N14N - discuss here?

(13)

$$\delta(O_2/N_2)_{\text{round-robin}} = \frac{1}{S_{O_2} \cdot X_{O_2}(1-X_{O_2})} \cdot \delta\tilde{r} - I_{CO_2} \cdot \Delta CO_2 - I_{Ar/N_2} \cdot \delta(Ar/N_2) - other interferences$$

where δτ is the difference in refractivity ratio τ = r(2537.27 Å)/r(4359.57 Å) between the round-robin cylinder and the SIO
reference, determined via interferometric comparisons with secondary reference gases linked to the primary suite. S02
=0.03397 is a constant sensitivity factor, X02 is the mole fraction of the SIO reference, IC02 is a constant (1.0919 per meg/ppm), and ΔCO2 is the difference in CO2 mole fraction from the SIO reference (363.29 µmol mol-1). SIO data are routinely corrected for CO2 interference. The apply additional corrections for Ar/N2, Ne, He, Kr, Xe, CH4, N2O, and CO. The additional corrections are effectively constant (or small) in natural air. They can usually be neglected in comparisons of natural air samples. However, these corrections cannot be neglected in relating the SIO scale to an absolute O2/N2 reference based on the round-robin cylinders, which addition relating the SIO scale to an absolute O2/N2 reference based on the round-robin cylinders, which addition relating the SIO scale to an absolute O2/N2 reference based on the round-robin cylinders, which are relevant in Eq. (13) as references for relative refractivity. Therefore, the exact Ar/N2 ratio and other gases' abundances in typical background air. Notably, the primary reference gases are relevant in Eq. (13) as references for relative refractivity.

Ar/N2 = 0.0119543, Ne/N2 = 2.328 × 10-5, He/N2 = 6.71×10-6, Kr/N2 = 1.46×10-6, Xe/N2 = 1.11×10-7, CH4 = 1.8 µmol mol-1, N2O = 0.3 µmol mol-1, CO = 0.1 µmol mol-1. Here, Ar/N2 is from Aoki et al. (2019), and the other (noble gas)/N2 ratios are from Glueckhauf (1951). The sensitivity S02 and interference factors (e.g., IAr/N2 = -0.0124) in Eq. (13) are based on refractivity data for the pure gases and natural air (Keeling, 1988, Keeling et al., 1998) using Xe data from Kronjäger (1936) (also see Keeling et al., 2020). The quantity δ(Ar/N2) was computed using the AIST gravimetric data, δ(Ar/N2) = 20 ((Ar/N2)grav/0.0119543 -1).

abundances of other gases in the SIO reference are not directly relevant. For background air, the following values were adopted:

The Ar/N2 interference  $(-I_{Ar/N_2} \cdot \delta(Ar/N_2))$  ranges from -55 to + 24 per meg, depending on the round-robin cylinder. The sum of the remaining interferences, other than for CO2 (- *other interferences*), is effectively constant at -14.3 per meg. The largest individual contributions are from Ne (-32.8 per meg) and CH4 (+11.9 per meg).

**25 3 Results and Discussion**

**3.1 Stability of $\delta(O_2/N_2)$ During Inter4 comparison**

The  $\delta(O_2/N_2)_{round-robin}$  values were measured four times using the mass spectrometer by EMRI/AIST to evaluate the stability of the  $O_2/N_2$  ratios of the standard mixtures during the inter-comparison experiment. The initial  $\delta(O_2/N_2)_{round-robin}$  values in the measurement of four times were used as the EMRI/AIST assigned values. The  $\delta(O_2/N_2)_{round-robin}$  values were calculated against

30 the EMRI/AIST scale. The EMRI/AIST scale's stability was evaluated by measuring the values of  $\delta(O_2/N_2)$  in three working

| TNumber: 1         | Author: stephens    | Subject: Cross-Out Date: 5/24/2021 11:01:36 AM |
|--------------------|---------------------|------------------------------------------------|
| SIO applies        |                     |                                                |
| TNumber: 2         | Author: stephens    | Subject: Cross-Out Date: 5/24/2021 11:02:44 AM |
|                    |                     |                                                |
| TNumber: 3         | Author: stephens    | Subject: Highlight Date: 5/24/2021 11:05:02 AM |
| consider moving up | o immediately after | So2, Xo2, and Ico2 are defined.                |
| TNumber: 4         | Author: stephens    | Subject: Cross-Out Date: 5/24/2021 11:26:22 AM |

---

## Author Comment (AC2)

RC1: 'Comment on amt-2020-481', Anonymous Referee #1, 21 Mar 2021

**General comments:**

Language: in some places the English writing could be improved, I have listed some suggestions in the detailed comments below.

Abstract: the abstract would benefit for having a number to represent the differences in precision between the laboratories, next to the offsets. The span offsets given in % are not directly clear, and it would be helpful if they could be expressed differently. It is also not clear from the abstract how this span offsets leads to a value of 0.29 PgC/yr in the carbon budget, and in which direction the shift is. It would be good to elaborate on this. It would be helpful to specify what the term zero offsets represents, so that the abstract is easier to read without reading the manuscript first. It would be good to include a quantification of the comparison of the 2 records at HAT in the final sentence. The "temporal drift" in line 21 could also be further explained, e.g. with details on the time period.

Response: The description about the offsets was revised in overall to be able to understand what the value is (p1, L25-L28). We simply described how to estimate a value of 0.29 PgC/yr.

But this value was update and revised to 0.30PgC/yr. The words of "zero offsets" was revised to "deviations in the measured $\delta$ (O2/N2) values on laboratories' scales" (p1, L28). A result of the comparison of the 2 records at HAT was added in the final sentence(p2, L1-L2). Time period was added to the "temporal drift"(p1, L23).

Page 3, lines 25-28: it is not fully clear to me how the correction for the expansion rates are applied and how these are measured.

Response: We cleared how the correction for the expansion rates are applied and how the rates are measured. (p4, L8-L12)

Page 4, line 10: it would be good to add information on the choice of the range, it seems quite a large range in comparison to observed values.

Response: We selected a large range in comparison to observed values in order to evaluate the difference of the respective span sensitivities accurately. We added the information on the choice of range. (p4, L24-L25)

Page 5, line 3: how are these values determined? (5000, 1000, 500 times).

Response: The values were calculated based on the abundances of $^{17}O^{17}O$ and $^{17}O^{16}O$, $^{18}O^{18}O$

and $^{18}O^{16}O$, and $^{15}N^{15}N$ and $^{15}N^{14}N$. We added to the sentence about the calculation. (P5,

L21)

Page 5, line 28: what is sufficiently small?

**Response:** We revised the sentence to "sufficiently smaller than those of $^{17}O^{16}O$, $^{18}O^{16}O$, and $^{15}N^{15}N$". (P6, L20-21)

Page 6, line 13: why are these values constant?

**Response:** We added why these values are constant. (P7, L6-L7)

Page 7, line 2: can the authors also provide the long-term stability for the other labs?

**Response:** Because the long-term stability of NIES and AIST already described, the stability of SIO and TU were added. (TU: P8, L5-L7, SIO: P8, L28-P9,1)

Page 7, line 27: it would be good to explain that the SIO scale is defined to be 0 per meg, because it is used internationally.

**Response:** We added the sentence of "of which scale is defined as $\delta (O_2/N_2) = 0$". (P8, L26)

Page 9, line 11: (and other places in the text): what does the "expanded uncertainty" represent?

**Response:** We explained the expanded uncertainty in text (P10, L14-L16).
Expanded uncertainty (U) was represented using standard uncertainty (u) and coverage factor (k)by the following equation,

   $U=ku$

We used the coverage factor of 2(k=2) which means $\approx$ a 95% level of confidence.

Page 9, line 14: this seems a large drift in a couple of years' time, so "slightly" might not be the appropriate word here. The explanations for the drift because of the oxidation inside the cylinders seems to be different for each cylinders, are the corrections made for each cylinder separately? Are these the regular cylinders, also used for maintaining the NMJI/AIST scale? Or is this only used for the round-robins?

**Response:** We removed "slightly" and explained that the corrections for each cylinder were performed separately (P10, L19, L28). We used cylinders having inner wall treated for storing O2 standard gases. NIES and EMRI/AIST also use same type of cylinders.

Page 9, line 32: Where do we see the long-term drift of each laboratory's scale?

**Response:** The sentence about the long-term stabilities of each laboratory's scale were added in section 2.3 (EMRI/AIST:p6, L8-L9, NIES:p7,L26-L27, TU: p8,L5-L7, SIO: p8, L28-p9, L1).

Page 10, line 10: what do these percentages represent?

Response: This percentages represent relative deviation from span sensitivity of the NMIJ/AIST scale. We revised the sentence. (P11, L19-20)

Page 10, line 12: could you elaborate on the filling years?

Response: We revised this sentence in overall. (P11, L23-L24)

Page 10, line 16: how are the results consistent with the GOLLUM program? Can this be quantified?

Response: Our results were consistent with those of the GOLLIM program within uncertainty. We add the words of "within their uncertainty". (P11, L28-L29)

Page 10, line 17: could you quantify "slightly bigger"?

Response: This sentence was removed. (P11, L29)

Page 10, line 22: can you quantify how this study shows that the labs can be compared?

Response: We revised "shows" to "aims". (P12, L5)

Page 10, line 25: how was it confirmed that the isotope ratios did not differ significantly?

Response: We add some sentences to get good understanding of this part. (P12, L8-L12)

Page 10, line 29: why not both against the SIO scale?

Response: We understand the SIO scale is internationally used. But purpose of this manuscript is the comparison between individual laboratories' scale values and gravimetric values directly. Therefore, we discussed NIES and EMRI/AIST scales based on NMIJ/AIST scale.

Page 11, line 2: how is the value of -6.6 per meg derived, and should it be compared to the goal of intercompatibility of 5 per meg?

Response:  We revised the sentence according your comments. (P12, L22-L25)

Page 11, lines 16-20: the GCB paper has been updated in the meantime twice, and it would be best to use the numbers from Friedlingstein et al. 2020. Line 18 does not seem to be a full sentence and it is not clear to me what the 0.29 PgC/yr correction is (e.g. from land to ocean, or the other way around?). Table 5: it would be good to add the numbers from Friedlingstein et al. 2020 in the table for reference.

**Response:** We updated our data using the numbers from Friedlingstein et al. 2020 and the sentence about the 0.29 PgC/yr correction was revised. (P13, L7-L8)

Page 11, line 30: "first time in the world": what is the first time in the world? The GOLLUM program is also an intercomparison program between laboratories.

**Response:** Because the span sensitivities of the respective laboratories are not compared in the GOLLUM program, quantifying respective span sensitivities is performed the first time in the world in this study.

Page 11, line 31: rewrite the 0.29 PgC/yr, to be more specific what the number means. See comment above.

**Response:** We revised the sentence in overall. (P13, L24-L25)

Page 12, line 3: what does "other four" mean in comparison to the GOLLUM program?

**Response:** We used five round-robin cylinders and "other four" mean four cylinders in the five cylinders. (P13, L27-L28)

Page 12, lines 4-6: it would be good to be more specific here on the implications, rather than repeating the causes for the decrease.

**Response:** We described the implications in the sentence according to your comment. (P13, L29-p14,L1)

Page 12, line 11: can you quantify the bias?

**Response:** We add the value of the bias. (P14, L6)

Page 12, lines 11-12: how do the results improve the carbon budget and OHC increase?

**Response:** We added the sentence about improvement of the carbon budget and OHC increase. (P14, L7-L8)

Conclusions: it would be useful to include an outlook. Will this intercomparison continue in the future? Will other laboratories be invited to participate?

**Response:** We added an outlook in end of the conclusions. (P14, L8-L11)

Table 1: can the authors add more information about the expansion rates? What is meant with the standard uncertainty?

Response: We add more information about the expansion rate. We explained that the standard uncertainty was calculated according to the law of propagation of uncertainties.

Figure 2: what is on the x axis?

Response: We revised the x axis which represent gravimetric values.

Table 3: how is the standard uncertainty determined?

Response: We added how the standard uncertainty is determined.

Figure 3: should the y-axis in panel a read NMJI/AIST instead of grav? Maybe also include the average residuals to compare lab precisions?

Response: This figure represents the relation of the gravimetric values and the measured values. We revised the caption.

Figure 4: maybe add a panel with the differences/bias? Why not on the Scripps scale? It would be good to include in the caption that these are duplicate samples, not measurements of the same flasks.

Response: The difference values were added in figure 4. We described that these are duplicate samples. We explained why not on the Scripps scale in previous part.

Table 5: change to Friedlingstein et al. 2020 (see comment above), and add numbers for comparison. Which numbers are Tohjima et al. 2019? Rewrite "changing rate".

Response: We changed values of fossil fuel and atmospheric $CO_2$ to Friedlingstein et al. 2020 and revised the footnotes.

Detailed comments:
Line 2: "molar fraction" could be changed to "mole fraction", which is more commonly used in the field, throughout the text.

Response: We understand that "mole fraction" is more commonly used in the field. But we should use "molar fraction" because derived quantities should be defined by quantities and not by units (mole is a unit). Angles can be defined as 'length ratios' and not as 'meter ratios'. A mass fraction is not called gram fraction either.

Line 3: explain "some" stoichiometric ratios.

**Response:** We revised from "some" to "distinct". (P2, L7)

Lines 2-5: references to earlier studies would be appropriate here.

**Response:** We added references of earlier studies. (P2, L6-L8)

Line 6: "changes" should be "measurements".

**Response:** "change in" was revised to "measurements of ". (P2, L10)

Line 11: rewrite "the ocean uptakes"

**Response:** We revised to "the ocean uptakes takes in more than 90% of the Earth's excess energy evaluated based on ocean temperature measurements using Argo floats" (P2, L15-17)

Line 14: "precision" -> "precise" and rewrite micro-mole-per-mole

**Response:** "precision" -> "precise" (P2, L18)

Line 27: "per meg" instead of "per meg units"

**Response:** we revised from "per meg" to "per meg units" (P3, L2)

Page 3

Line 6: "remain"?

**Response:** We revised the sentence. (P3, L16)

Line 22: explain "round-robin"

**Response:** We add the explanation of round-robin in introduction. (P3, L25-26)

Page 4

Line 6: "hereafter the" -> "hereafter called the"

**Response:** we revised from "hereafter the" to "hereafter called the" (P4, L20-L21)

Line 24: why are the Ar values much more variable compared to tropospheric air?

**Response:** We revised the sentence because the "variable" is wrong. (P5, L10)

Page 7

Line 10: what is meant with "unequal to"?

**Response**: We revised it to "not equivalent to". (P8, L9)

Page 8

Lines 5-8: the sensitivity factor and interference factors could be further explained to be clearer.

**Response**: We explained the sensitivity factor and interference. (P9, L11-12)

Page 9

Line 1: what are "changing rates"?

**Response**: The changing rates represent change speed of $\delta$ (O$_2$/N$_2$). We revised the sentence a little. (P10, L7-8)

Lines 29-30: what is meant here by selecting mixtures from the round-robins?

**Response**: We removed the sentence. (P11, L5)

Page 10:

Line 5: rewrite "Figure 3a plots"

**Response**: We revised to "Figure 3a represents". (P11, L14)

Page 11

Line 10: "corrected" -> "improved"?

**Response**: We revised from "corrected" to "improved". (P12, L29)

Line 13: rewrite "secular changing rate"

**Response**: We revised to the average changing rate of atmospheric O$_2$/N$_2$ ratio and CO$_2$ molar fraction reported on the NIES scale. (P13, L3)

---

## Author Comment (AC3)

**RC2**: 'Comment on amt-2020-481', Britton Stephens, 24 May 2021

**Page: 1**

Number: 1 Subject: Highlight, intercomparison is a word on its own (no hyphen needed)

**Response:** We removed hyphen from "inter-comparison" through the text.

Number: 2 Subject: Highlight, "on a Round-Robin Exercise" might be more informative / grammatically correct

**Response:** We revised the sentence according to your comment. (P1, L2)

Number: 3 Subject: Cross-Out, a

**Response:** We revised the sentence according to your comment. (P1,L26)

Number: 4 Subject: Inserted Text, global

**Response:** We revised the sentence according to your comment. (P1,L27)

Number: 5 Subject: Cross-Out, based on trends in atmospheric CO2 and d(O2/N2)

**Response:** We added "based on trends in atmospheric $CO_2$ and $\delta(O_2/N_2)$" (P1, L27)

**Page: 2**

Number: 1 Subject: Inserted Text distinct

**Response:** We revised the sentence according to your comment. (P2, L7)

Number: 2 Subject: Inserted Text, measurements of

**Response:** We revised the sentence according to your comment. (P2, L10)

Number: 3 Subject: Inserted Text, E

**Response:** We revised the sentence according to your comment. (P2, L15)

Number: 4 Subject: Inserted Text, as

**Response:** We revised the word from "and is" to "as" (P2, L15)

Number: 5 Subject: Inserted Text, s

**Response:** We revised the sentence according to your comment. (P2, L16)

Number: 6 Subject: Inserted Text, e

**Response:** We revised the sentence according to your comment. (P2, L18)

Number: 7 Subject: Inserted Text, a

**Response:** We revised the sentence according to your comment. (P2, L24)

Number: 8 Subject: Highlight, cite Aoki et al., AMT 2019 instead / also

Author: stephens Subject: Sticky Note Date:

given global trends, it would also be helpful to give a year (1967-70?) corresponding to this mole fraction estimate

**Response:** We cited "Aoki et al., AMT 2019" and added a year corresponding to the molar fractions. (P3, L2-3)

Number: 9 Subject: Highlight, previous sentence says this is calibrated, so "calibrated" or "measured" would be better there than "assumed"

**Response:** We revised the sentence according to your comment. (P3, L8)

Number: 10 Subject: Inserted Text, of

**Response:** We revised the sentence according to your comment. (P3, L8)

Number: 11 Subject: Cross-Out, a span stability

**Response:** We revised the sentence according to your comment. (P3, L9)

**Page: 3**

Number: 1 Subject: Inserted Text, stability

**Response:** We revised the sentence according to your comment. (P3, L10)

Number: 2 Subject: Inserted Text, in absolute terms

Author: Subject: Sticky Note, a bit tricky to talk about absolute uncertainty in per meg (a relative unit) - perhaps give in mole fraction instead?

**Response:** The sentence was revised as expressed as the mole fraction to understand it in absolute terms. But we also add corresponding value in per meg unit. (P3, L14)

Number: 3 Subject: Inserted Text

**Response:** We didn't know where to revise. (P3, L15)

Number: 4 Subject: Cross-Out:

**Response:** We revised the sentence according to your comment. (P3, L22)

Number: 5 Subject: Highlight

Here, it would be good to point out (as you do below) that by comparing to gravimetric mixtures prepared over time, the trend uncertainty can be addressed. Just checking against them once can fix span biases, but not necessarily zero drift over time.

**Response:** How to evaluate span bias and zero drift is different. We described separately advantages with respect to span bias and zero drift obtained by comparing the scales with gravimetric mixtures. (P3, L20-22)

Number: 6 Subject: Inserted Text, the

**Response:** We revised the sentence according to your comment. (P3, L23)

Number: 7 Subject: Highlight, give materials?

**Response:** We revised the sentence according to your comment. (P4, L6)

Number: 8 Subject: Inserted Text, . They

**Response:** We revised the sentence according to your comment. (P4 L6)

Number: 9 Subject: Inserted Text

**Response:** We revised it in the overall because the sentence was difficult to understand. (P4 L9)      .

**Page: 4**

Number: 1 Subject: Highlight, Why not use a more modern O2 mole fraction estimate? If it does not matter, consider saying so. Since you have gravimetric determinations, I'm guessing the assignment of zero on the scale is arbitrary, so it might help to say "we arbitrarily assign zero on the NMIJ/AIST scale to correspond to a ratio of 0.26825" and also maybe that this corresponds to the late 1960s.

**Response:** We adopted a more modern O2 mole fraction estimate according to your comment. (P4, L22-23)

Number: 2 Subject: Cross-Out Date:

**Response:** We revised the sentence according to your comment. (P4 L27)

Number: 3 Subject: Cross-Out Date:

**Response:** We revised the sentence according to your comment. (P4 L28)

Number: 4 Subject: Inserted Text, as

**Response:** We revised the sentence according to your comment. (P5 L1)

Number: 5 Subject: Highlight, this is unclear - does it refer to the standard, the true value of the standard, or the reported values by the labs? Is the "round-robin" subscript necessary or could it be removed throughout?

**Response:** The $\delta(O2/N2)_{round-robin}$ represent the measured values by the labs. We adopted the subscript to distinguish the round-robin values and other values clearly. This sentence was revised according your comment. (P5 L2)

Number: 6 Subject: Inserted Text, m

**Response:** We didn't know where to revise. (P5 L4)

Number: 7 Subject: Cross-Out

**Response:** We revised the sentence according to your comment. (P5 L9)

Number: 8 Subject: Highlight, considering giving values on established reference scales

**Response:** We added the sentence according to your comment. (P5 L11-14)

**Page: 5**

Number: 1 Subject: Highlight, might be helpful to start by saying why these corrections are needed - dilution, peak overlap, etc.

**Response:** we explained the reason that these corrections are needed. (P5 L22-24)

Number: 2 Subject: Inserted Text, for (first sentence says these are O2/N2 corrections, not Ar corrections)

**Response:** Because we corrected O2/N2 using the deviation in Ar molar fraction from atmospheric value, the sentence was revised. (P5 L29)

Number: 3 Subject: Inserted Text, for (first sentence says thesea are O2/N2 corrections, not Ar corrections)

**Response:** because we corrected O2/N2 using the deviation in Ar molar fraction from atmospheric value, the sentence was revised. (P6 L1)

Number: 4 Subject: Highlight, say 'corrected' here instead? (first sentence already says 'calculated based on' 16/14)

**Response:** We revised the sentence according to your comment. (P6 L15)

**Page: 6**

Number: 1 Subject: Inserted Text, nearly (?)

**Response:** We revised the sentence in overall. (P7 L6-7)

Number: 2 Subject: Highlight, consider rewording for clarity - if you already have gravimetric Ar and N2, why do you need (O2+Ar)/N2 and CO2 to get (Ar/N2)? Also, change "in this study" to "For NIES" or equivalent.

**Response:** We changed the values to gravimetric Ar and N2. NIES's $\delta(O2/N2)$ values was recalculated using the gravimetric values. (P7,L24-25)

**Page: 7**

Number: 1 Subject: Inserted Text, cylinders

**Response:** We revised the sentence according to your comment. (P7 L26)

Number: 2 Subject: Inserted Text, with respect to these cylinders (primaries could still drift right?)

**Response:** We revised the sentence according to your comment. (P7 L27)

Number: 3 Subject: Highlight, might be of interest to say why 15N14N is used instead of 14N14N

Subject: Sticky Note Date: 5/24/2021 11:21:18 AM

more specifically, 2 sentences later says mass spec measures 28 - why is it not used?

**Response:** We used mass 29 because the spread of both ion beams for mass 28 and 32 was too wide to measure simultaneously. This sentence was revised according your comment. (P8, L7-9)

Number: 4 Subject: Highlight, this is of course true because they are on different scales - did you instead mean d(O2/N2)round-robin here?

**Response:** The word "unequal" was wrong. We revised it to "not equivalent"(P8, L9).

Number: 5 Subject: Highlight, Table 2 mentions mass-spec measurements of 40Ar and 14N14N - discuss here?

**Response:** Mass-spec measurements of 40Ar and 14N14N were removed from Table 2 because SIO used gravimetric values instead of mass-spec measurements of 40Ar and 14N14N.

**Page: 8**

Number: 1 Subject: Cross-Out, SIO applies

**Response:** We revised the sentence according to your comment. (P9, L12)

Number: 2 Subject: Cross-Out

**Response:** We revised the sentence according to your comment. (P9, L15)

Number: 3 Subject: Highlight, consider moving up immediately after So2, Xo2, and Ico2 are defined.

**Response:** We moved up it immediately after $S_{o2}$, $X_{o2}$, and $I_{co2}$ are defined. (P9, L11-12)

Number: 4 Subject: Cross-Out,

**Response:** We revised the sentence according to your comment. (P10, L2)

**Page: 9**

Number: 1 Subject: Cross-Out, (unclear what is averaged)

**Response:** We revised the sentence in overall. (P10, L7-8)

Number: 2 Subject: Cross-Out

**Response:** We revised the sentence according to your comment. (P10, L16)

Number: 3 Subject: Inserted Text, '

**Response:** We revised the sentence according to your comment. (P10, L16)

Number: 4 Subject: Highlight, this would increase d($O_2/N_2$), no?

**Response:** Because this sentence was wrong, we revised it. (P10, L24-25)

Number: 5 Subject: Cross-Out, EMRI/AIST

**Response:** This word is correct, while the words of data is wrong. We revised the word from data to date. (P10, L27)

Number: 6 Subject: Inserted Text, at EMRI/AIST?

**Response:** We revised the sentence according to your comment. (P11, L1)

Number: 7 Subject: Highlight, Not sure what this means - selected for what?

**Response:** We removed the sentence. (P11, L5)

Number: 8 Subject: Inserted Text, which showed

**Response:** We revised the sentence according to your comment. (P11, L6)

**Page: 10**

Number: 1 Subject: Inserted Text, Their

**Response:** We revised the sentence according to your comment. (P11, L10)

Number: 2 Subject: Inserted Text,

**Response:** We revised the sentence according to your comment (This overlaps Number: 1?). (P11, L10)

Number: 3 Subject: Inserted Text, values.

**Response:** We revised the sentence according to your comment. (P11, L11)

Number: 4 Subject: Cross-Out

**Response:** We revised the sentence according to your comment. (P11, L12)

Number: 5 Subject: Cross-Out

**Response:** We revised the sentence according to your comment. (P11, L12)

Number: 6 Subject: Highlight, Table 4 calls these "standard uncertainties" - say how calculated, and if standard

deviations, say of what.

**Response:** "standard deviations" is wrong and it is "standard uncertainties". We added "the standard uncertainties which were calculated based on the Deming least-square fit". (P11, L22-23)

Number: 7 Subject: Cross-Out

**Response:** We revised the sentence according to your comment. (P11, L27)

Number: 8 Subject: Cross-Out

**Response:** We revised the sentence according to your comment. (P11, L27)

Number: 9 Subject: Highlight, see alternate suggestion of how to cite (it is not clear here that "the GOLLUM comparison" is a citation, and some more description is warranted). Should probably also cite: WMO, 2005: Global Atmosphere Watch, 12th WMO/IAEA Meeting of Expert on Carbon Dioxide Concentration and Related Tracers Measurements Techniques (Toronto, Canada, 15-18 September 2003). GAW Report No.161, WMO TD No. 1275, Geneva. and/or A. Manning pers. comm. for the actual values since they are not included in either of these.

**Response:** We revised how to cite according to your suggestion. (P11, L27-29)

Number: 10 Subject: Inserted Text, a

**Response:** We revised the sentence according to your comment. (P11, L27)

Number: 11 Subject: Inserted Text , the GOLLUM exercise coordinated by SIO and the University of East Anglia from 2003-2014 (GOLLUM, 2015),

**Response:** We revised the sentence according to your comment. (P11, L27-29)

Number: 12 Subject: Cross-Out

**Response:** We revised the sentence according to your comment. (P12, L5)

**Page: 11**

Number: 1 Subject: Inserted Text, )

**Response:** We revised the sentence according to your comment. (P12, L15-16)

Number: 2 Subject: Inserted Text, We found a change of

**Response:** We revised the sentence according to your comment. (P13, L9-10)

Number: 3 Subject: Inserted Text, to

**Response:** We revised the sentence according to your comment. (P13, L9)

Number: 4 Subject: Inserted Text,

**Response:** We didn't know where to revise.

Number: 5 Subject: Inserted Text, are

**Response:** We revised the sentence according to your comment. (P13, L11)

Number: 6 Subject: Inserted Text, to

**Response:** We revised the sentence according to your comment. (P13, L11)

Number: 7 Subject: Cross-Out

**Response:** We revised the sentence according to your comment. (P13, L12)

Number: 8 Subject: Cross-Out

**Response:** We revised the sentence according to your comment. (P13, L16)

Number: 9 Subject: Highlight, to support this statement, please say what value Resplandy used for their estimate of span uncertainty (2% 1-sigma) and what value(s) you would recommend instead. For example, would you recommend 1.6% (standard deviation of slopes reported here) without any correction, and some smaller uncertainty if a span correction is applied?

**Response**: We added what value they used for their estimate of span uncertainty and what value(s) we recommend instead of their estimate of span uncertainty.(P13,L15-18)

Number: 10 Subject: Cross-Out

**Response:** We revised the sentence according to your comment. (P13, L20)

Number: 11 Subject: Cross-Out

**Response:** We revised the sentence according to your comment. (P13, L24)

Number: 12 Subject: Inserted Text, changes

**Response:** We revised the sentence in overall. (P13, L24-25)

**Page: 12**

Number: 1 Subject: Cross-Out, We speculate that t

**Response:** We revised the sentence in overall. (P13, L28-30)

Number: 2 Subject: Cross-Out

**Response:** We revised the sentence in overall. (P13, L29-P14,L1)

Number: 3 Subject: Highlight, adsorption?

**Response:** We revised the sentence in overall. (P13, L29-P14,L1)

Number: 4 Subject: Inserted Text, ,

**Response:** We revised the sentence in overall. (P13, L29-P14,L1)

Number: 5 Subject: Inserted Text, than t

**Response:** We revised the sentence in overall. (P13, L29-P14,L1)

Number: 6 Subject: Inserted Text, from

**Response:** We revised the sentence in overall. (P13, L29-P14,L1)

Number: 7 Subject: Cross-Out

**Response:** We revised the sentence in overall. (P14, L4-5)

Number: 8 Subject: Highlight, uncertainty? bias was within uncertainty, no?

**Response:** Because values of NIES and AIST by the conversion was consistent within uncertainty. We added "within uncertainty" to the sentence. (P14, L4-6)

Number: 9 Subject: Highlight, I suggest acknowledging Andrew Manning for the GOLLUM results.

**Response:** We were grateful to Manning for the GOLLUM results in acknowledgments. (P14, L14-15)

**Page: 13**

Number: 1 Subject: Highlight, This link does not work. This one does: https://gollum.uea.ac.uk/apo-2015.shtml

**Response:** the link was revised to "https://gollum.uea.ac.uk/apo-2015.shtml" (P16, L6)

**Page: 17(Table 1)**

Number: 1 Subject: Highlight, formatting issue here

**Response:** We revised the sentence according to your comment.

Number: 2 Subject: Inserted Text, dry air?

**Response:** We described that figures are given in the unit of $\mu mol\ mol^{-1}$ in dry air. (P20, L10)

Number: 3 Subject: Highlight, see earlier comments - give reference year and consider using more recent value

**Response:** The sentence was revised according to your comment (P20,L11-L13).

**Page: 18 (Table2)**

Number: 1 Subject: Highlight, line spacing in this column could be improved

**Response:** line spacing in this column was improved

**Page: 19 (Figure 1)**

Number: 1 Subject: Inserted Text, as

**Response:** We revised the sentence according to your comment.

**Page: 20 (Figure2)**

Number: 1 Subject: Highlight, should y-axis label on 2a be the same as on 3a?

**Response:** We revised y-axis label of the Figure 2a according to your comment.

Number: 2 Subject: Inserted Text, at EMRI/AIST

**Response:** We revised the sentence according to your comment.

Number: 3 Subject: Highlight, say what whiskers are here

**Response:** The whiskers were explained according to your comment.

**Page: 21 (Table 3)**

Number: 1 Subject: Highlight, specify/remind here or in footnote that these are all reported on different (lab-specific)

scales

**Response:** We specified in footnote that these are all reported on own scales

**Page: 22 (Figure3)**

Number: 1 Subject: Highlight, say what whiskers are

**Response:** The whiskers were explained according to your comment.

**Page: 23 (Table4)**

Number: 1 Subject: Highlight, This term is new here - say what "GOLLUM 15" refers to. I suggest just saying "GOLLUM"

as the 15 doesn't really indicate anything about the experiment (which ran 2003-2014) other than the date of the cited

presentation.

**Response:** We revised the sentence according to your comment.

Number: 2 Subject: Inserted Text

**Response:** We didn't know where to revise.

Number: 3 Subject: Inserted Text, from the Deming fit. (true?)

**Response:** We revised the sentence according to your comment.

Number: 4 Subject: Inserted Text

**Response:** We didn't know where to revise.

Number: 5 Subject: Inserted Text, the

**Response:** We revised the sentence according to your comment.

Number: 6 Subject: Inserted Text, the individual laboratory

**Response:** We revised the sentence according to your comment.

Number: 7 Subject: Highlight, "provided by Andrew Manning" instead? Or did you internally redo this comparison

based on SIO, TU, and NIES results?

**Response:** We revised the sentence according to your comment.

**Page: 25 (Figure5)**

Number: 1 Subject: Highlight, panel letters missing

**Response:** We revised the panel letters missing

Number: 2 Subject: Highlight, y-axes in (b) reflect conversion to NMIJ/AIST?

**Response:** The y-axes in (b) was updated because we assigned $0.2093391/ 0.7808943 = 0.2680761$ as $\delta$ (O2/N2)

$_{NMIJ/AIST}$=0. It reflects conversion to NMIJ/AIST

---

## Author Response (AR2)

1   Suggestions for technical corrections:

2   L26: ⋯. corresponded TO ⋯. (not with)

3   Response: We revised the sentence according to the comment.

4

5   L93: The details of the gravimetric preparation technique were given ⋯⋯

6   Response: We revised the sentence according to the comment.

7

8   L97: ⋯ acting ON a ⋯

9   Response: We revised the sentence according to the comment.

10

11  L322: I suggest rewording to: The goal of this study is to make the observational data from

12  different laboratories directly comparable.

13  Response: We revised the sentence according to the comment.

14

15  L370/71: delete "in the world"

16  Response: We revised the sentence according to the comment.

17

18  L385: ⋯ estimation of THE atmospheric ⋯

19  Response: We revised the sentence according to the comment.